# Inducing Equilibria via Incentives: Simultaneous Design-and-Play Ensures Global Convergence

**Boyi Liu**[*], **Jiayang Li**[*]
Northwestern University
{boyiliu2018,jiayangli2024}@u.northwestern.edu

**Zhuoran Yang**
Yale University
zhuoran.yang@yale.edu

**Hoi-To Wai**
The Chinese University of Hong Kong
htwai@se.cuhk.edu.hk

**Mingyi Hong**
University of Minnesota
mhong@umn.edu

**Yu (Marco) Nie,    Zhaoran Wang**
Northwestern University
{y-nie,zhaoran.wang}@northwestern.edu

## Abstract

To regulate a social system comprised of self-interested agents, economic incentives are often required to induce a desirable outcome. This incentive design problem naturally possesses a bilevel structure, in which a designer modifies the rewards of the agents with incentives while anticipating the response of the agents, who play a non-cooperative game that converges to an equilibrium. The existing bilevel optimization algorithms raise a dilemma when applied to this problem: anticipating how incentives affect the agents at equilibrium requires solving the equilibrium problem repeatedly, which is computationally inefficient; bypassing the time-consuming step of equilibrium-finding can reduce the computational cost, but may lead the designer to a sub-optimal solution. To address such a dilemma, we propose a method that tackles the designer's and agents' problems simultaneously in a single loop. Specifically, at each iteration, both the designer and the agents only move one step. Nevertheless, we allow the designer to gradually learn the overall influence of the incentives on the agents, which guarantees optimality after convergence. The convergence rate of the proposed scheme is also established for a broad class of games.

## 1 Introduction

A common thread in human history is how to "properly" regulate a social system comprised of self-interested individuals. In a laissez-faire economy, for example, the competitive market itself is the primary regulatory mechanism [47, 16]. However, a laissez-faire economy may falter due to the existence of significant "externalities" [8, 18], which may arise wherever the self-interested agents do not bear the external cost of their behaviors in the entirety. The right response, many argue, is to introduce corrective policies in the form of economic incentives (e.g., tolls, taxes, and subsidies) [32]. By modifying the rewards of the agents, these incentives can encourage (discourage) the agents to engage in activities that create positive (negative) side effects for the society, and thus guide the self-interests of the agents towards a socially desirable end. For example, carbon taxes can be levied on carbon emissions to protect the environment during the production of goods and services [35].

---

[*]Equal contribution.

36th Conference on Neural Information Processing Systems (NeurIPS 2022).

Surge pricing has been widely used to boost supply and dampen demand in volatile ride-hail markets [37]. Lately, subsidies and penalties were both introduced to overcome vaccine hesitancy in the world's hard-fought battle against the COVID-19 pandemic.

The goal of this paper is to develop a provably efficient method for guiding the agents in a non-cooperative game towards a socially desirable outcome — e.g., the one that maximizes the social welfare — by modifying their payoffs with incentives. The resulting problem may be naturally interpreted as a Stackelberg game [50] in which the "incentive designer" is the leader while the agents being regulated are the followers. Hence, it naturally possesses a bilevel structure [3]: at the upper level, the "designer" optimizes the incentives by anticipating and regulating the best response of the agents, who play a non-cooperative game at the lower level. As the lower-level agents pursue their self-interests freely, their best response can be predicted by the Nash equilibrium [39], which dictates no agent can do better by unilaterally changing their strategy. Accordingly, the incentive design problem is a mathematical program with equilibrium constraints (MPEC) [30].

In the optimization literature, MPECs are well-known for their intractability [10]. Specifically, even getting a first-order derivative through their bilevel structure is a challenge. In the incentive design problem, for example, to calculate the gradient of the designer's objective at equilibrium, which provides a principled direction for the designer to update the incentives, one must *anticipate* how the equilibrium is affected by the changes [17]. This is usually achieved by performing a sensitivity analysis, which in turn requires differentiation through the lower-level equilibrium problem, either implicitly or explicitly [25]. No matter how the sensitivity analysis is carried out, the equilibrium problem must be *solved* before updating the incentives. The resulting algorithm thus admits a double loop structure: in the outer loop, the designer iteratively moves along the gradient; but to find the gradient, it must allow the lower level game dynamics to run its course to arrive at the equilibrium given the current incentives.

Because of the inherent inefficiency of the double-loop structure, many heuristics methods have also been developed for bilevel programs in machine learning [27, 29, 14]. When applied to the incentive design problem, these methods assume that the designer *does not solve* the equilibrium exactly to evaluate the gradient. Instead, at each iteration, the game is allowed to run just a few rounds, enough for the designer to obtain a reasonable approximation of the gradient. Although such a method promises to reduce the computational cost significantly at each iteration, it may never converge to the same optimal solution obtained without the approximation.

**Contribution.** In a nutshell, correctly anticipating how incentives affect the agents at equilibrium requires solving the equilibrium problem repeatedly, which is computationally inefficient. On the other hand, simply bypassing the time-consuming step of equilibrium-finding may lead the designer to a sub-optimal solution. This dilemma prompts the following question that motivates this study: *can we obtain the optimal solution to an incentive design problem without repeatedly solving the equilibrium problem?*

In this paper, we propose an efficient principled method that tackles the designer's problem and agents' problem simultaneously in a single loop. At the lower level, we use the *mirror descent* method [40] to model the process through which the agents move towards equilibrium. At the upper level, we use the *gradient descent* method to update the incentives towards optimality. At each iteration, both the designer and the agents only move one step based on the first-order information. However, as discussed before, the upper gradient relies on the corresponding lower equilibrium, which is not available in the single-loop update. Hence, we propose to use the implicit differentiation formula—with equilibrium strategy replaced by the current strategy—to estimate the upper gradient, which might be biased at the beginning. Nevertheless, we prove that if we improve the lower-level solution with larger step sizes, the upper-level and lower-level problems may converge simultaneously at a fast rate. The proposed scheme hence guarantees optimality because it can anticipate the overall influence of the incentives on the agents eventually after convergence.

**Organization.** In Section 2, we discuss related work. In Section 3, we provide the mathematical formulation of the incentive design problem. In Section 4, we design algorithms for solving the problem. In Section 5, we establish conditions under which the proposed scheme globally converges to the optimal solution and analyze the convergence rate. The convergence analysis is restricted to games with a unique equilibrium. In Section 6, we discuss how to apply our algorithms to games with multiple equilibria. Eventually, we conduct experiments to test our algorithms in Section 7.

**Notation.** We denote $\langle \cdot, \cdot \rangle$ as the inner product in vector spaces. For a vector form $a = (a^i)$, we denote $a^{-i} = (a^j)_{j \neq i}$. For a finite set $\mathcal{X} \in \mathbb{R}^n$, we denote $\Delta(\mathcal{X}) = \{\pi \in \mathbb{R}^n_+ : \sum_{x^i \in \mathcal{X}} \pi_{x^i} = 1\}$. For any vector norm $\| \cdot \|$, we denote $\| \cdot \|_* = \sup_{\|z\| \leq 1} \langle \cdot, z \rangle$ as its dual norm. We refer readers to Appendix A for a collection of frequently used problem-specific notations.

## 2 Related work

The incentive design problem studied in this paper is a special case of mathematical programs with equilibrium constraints (MPEC) [19], a class of optimization problems constrained by equilibrium conditions. MPEC is closely related to bilevel programs [10], which bind two mathematical programs together by treating one program as part of the constraints for the other.

**Bilevel Programming.** In the optimization literature, bilevel programming was first introduced to tackle resource allocation problems [7] and has since found applications in such diverse topics as revenue management, network design, traffic control, and energy systems. In the past decade, researchers have discovered numerous applications of bilevel programming in machine learning, including meta-learning (ML) [14], adversarial learning [22], hyperparameter optimization, [31] and neural architecture search [27]. These newly found bilevel programs in ML are often solved by gradient descent methods, which require differentiating through the (usually unconstrained) lower-level optimization problem [28]. The differentiation can be carried out either implicitly on the optimality conditions as in the conventional sensitivity analysis [see e.g., 2, 43, 4], or explicitly by unrolling the numerical procedure used to solve the lower-level problem [see e.g., 31, 15]. In the explicit approach, one may "partially" unroll the solution procedure (i.e., stop after just a few rounds, or even only one round) to reduce the computational cost. Although this popular heuristic has delivered satisfactory performance on many practical tasks [29, 36, 14, 27], it cannot guarantee optimality for bilevel programs under the general setting, as it cannot derive the accurate upper-level gradient at each iteration [53].

**MPEC.** Unlike bilevel programs, MPEC is relatively under-explored in the ML literature so far. Recently, Li et al. [25] extended the explicit differentiation method for bilevel programs to MPECs. Their algorithm unrolls an iterative projection algorithm for solving the lower-level problem, which is formulated as a variational inequality (VI) problem. Leveraging the recent advance in differentiable programming [2], they embedded each projection iteration as a differentiable layer in a computational graph, and accordingly, transform the explicit differentiation as standard backpropagation through the graph. The algorithm proposed by Li et al. [26] has a similar overall structure, but theirs casts the lower-level solution process as the imitative logit dynamics [6] drawn from the evolutionary game theory, which can be more efficiently unrolled. Although backpropagation is efficient, constructing and storing such a graph — with potentially a large number of projection layers needed to find a good solution to the lower-level problem — is still demanding. To reduce this burden, partially unrolling the iterative projection algorithm is a solution. Yet, it still cannot guarantee optimality for MPECs due to the same reason as for bilevel programs.

The simultaneous design-and-play approach is proposed to address this dilemma. Our approach follows the algorithm of Hong et al. [21] and Chen et al. [9], which solves bilevel programs via single-loop update. Importantly, they solve both the upper- and the lower-level problem using a gradient descent algorithm and establish the relationship between the convergence rate of the single-loop algorithm and the step size used in gradient descent. However, their algorithms are limited to the cases where the lower-level optimization problem is *unconstrained*. Our work extends these single-loop algorithms to MPECs that have an equilibrium problem at the lower level. We choose mirror descent as the solution method to the lower-level problem because of its broad applicability to optimization problems with constraints [40] and generality in the behavioral interpretation of games [34, 23]. We show that the convergence of the proposed simultaneous design-and-play approach relies on the setting of the step size for both the upper- and lower-level updates, a finding that echos the key result in [21]. We first give the convergence rate under mirror descent and the unconstrained assumption and then extend the result to the constrained case. For the latter, we show that convergence cannot be guaranteed if the lower-level solution gets too close to the boundary early in the simultaneous solution process. To avoid this trap, the standard mirror descent method is revised to carefully steer the lower-level solution away from the boundary.

# 3 Problem Formulation

We study incentive design in both atomic games [39] and nonatomic games [45], classified depending on whether the set of agents is endowed with an atomic or a nonatomic measure. In social systems, both types of games can be useful, although the application context varies. Atomic games are typically employed when each agent has a non-trivial influence on the rewards of other agents. In a nonatomic game, on the contrary, a single agent's influence is negligible and the reward could only be affected by the collective behavior of agents.

**Atomic Game.** Consider a game played by a finite set of agents $\mathcal{N} = \{1, \ldots, n\}$, where each agent $i \in \mathcal{N}$ selects a strategy $a^i \in \mathcal{A}^i \subseteq \mathbb{R}^{d^i}$ to maximize the reward received, which is determined by a continuously differentiable function $u^i : \mathcal{A} = \prod_{i \in \mathcal{N}} \mathcal{A}^i \to \mathbb{R}$. Formally, a joint strategy $a_* \in \mathcal{A}$ is a Nash equilibrium if

$$u^i(a_*^i, a_*^{-i}) \geq u^i(a^i, a_*^{-i}), \quad \forall \, a^i \in \mathcal{A}^i, \quad \forall i \in \mathcal{N}.$$

Suppose that for all $i \in \mathcal{N}$, the strategy set $\mathcal{A}^i$ is closed and convex, and the reward function $u^i$ is convex in $a^i$, then $a_* \in \mathcal{A}$ is a Nash equilibrium if and only if there exists $\lambda^i, \ldots, \lambda^n > 0$ such that [46]

$$\sum_{i=1}^{n} \lambda^i \cdot \left\langle \nabla_{a^i} u^i(a_*), a^i - a_*^i \right\rangle \leq 0, \quad \text{for all } a \in \mathcal{A}. \tag{3.1}$$

**Example 3.1** (Oligopoly model). In an oligopoly model, there is finite set $\mathcal{N} = \{1, \ldots, n\}$ of firms, each of which supplies the market with a quantity $a^i$ ($a^i \geq 0$) of goods. Under this setting, we have $\mathcal{A} = \mathbb{R}_+^n$. The good is then priced as $p(q) = p_0 - \gamma \cdot q$, where $p_0, \gamma > 0$ and $q = \sum_{j \in \mathcal{N}} a^i$ it the total output. The profit and the marginal profit of the firm $i$ are then given by

$$u^i(a) = a^i \cdot \left( p_0 - \gamma \cdot \sum_{j \in \mathcal{N}} a^j \right) - c^i, \quad \nabla_{a^i} u^i(a) = p_0 - \gamma \cdot \left( a^i + \sum_{j \in \mathcal{N}} a^j \right),$$

respectively, where $c^i$ is the constant marginal cost[2] for firm $i$.

**Nonatomic Game.** Consider a game played by a continuous set of agents, which can be divided into a finite set of classes $\mathcal{N} = \{1, \ldots, n\}$. We assume that each $i \in \mathcal{N}$ represents a class of infinitesimal and homogeneous agents sharing the finite strategy set $\mathcal{A}^i$ with $|\mathcal{A}^i| = d^i$. The mass distribution for the class $i$ is defined as a vector $q^i \in \Delta(\mathcal{A}^i)$ that gives the proportion of agents using each strategy. Let the cost of an agent in class $i$ to select a strategy $a \in \mathcal{A}^i$ given $q = (q^1, \ldots, q^n)$ be $c^{ia}(q)$. Formally, a joint mass distribution $q \in \Delta(\mathcal{A}) = \prod_{i \in \mathcal{N}} \Delta(\mathcal{A}^i)$ is a Nash equilibrium, also known as a Wardrop equilibrium [51], if for all $i \in \mathcal{N}$, there exists $b^i$ such that

$$\begin{cases} c^{ia}(q_*) = b^i, & \text{if } q_*^{ia} > 0, \\ c^{ia}(q_*) \geq b^i, & \text{if } q_*^{ia} = 0. \end{cases}$$

The following result extends the VI formulation to Nash equilibrium in nonatomic game: denote $c^i(q) = (c^{ia}(q))_{a \in \mathcal{A}^i}$, then $q_*$ is a Nash equilibrium if and only if [11]

$$\sum_{i \in \mathcal{N}} \lambda^i \cdot \left\langle c^i(q_*), q^i - q_*^i \right\rangle \geq 0, \quad \text{for all } q \in \Delta(\mathcal{A}). \tag{3.2}$$

**Example 3.2** (Routing game). Consider a set of agents traveling from source nodes to sink nodes in a directed graph with nodes $\mathcal{V}$ and edges $\mathcal{E}$. Denote $\mathcal{N} \subseteq \mathcal{V} \times \mathcal{V}$ as the set of source-sink pairs, $\mathcal{A}^i \subseteq 2^{\mathcal{E}}$ as the set of paths connecting $i \in \mathcal{N}$ and $\mathcal{E}^{ia} \subseteq \mathcal{E}$ as the set of all edges on the path $a \in \mathcal{A}^i$. Suppose that each source-sink pair $i \in \mathcal{N}$ is associated with $\rho^i$ nonatomic agents aiming to choose a route from $\mathcal{A}^i$ to minimize the total cost incurred. Let $q^{ia} \in \Delta(\mathcal{A}^i)$ be the proportion of travelers using the path $a \in \mathcal{A}_w$, $x_e \in \mathbb{R}_+$ be the number of travelers using the edge $e$ and $t^e(x^e) \in \mathbb{R}_+$ be the cost for using edge $e$. Then we have $x^e = \sum_{i \in \mathcal{N}} \sum_{a \in \mathcal{A}^i} \rho^i \cdot q^{ia} \cdot \delta^{eia}$, where $\delta^{eik}$ equals 1 if $e \in \mathcal{E}^{ia}$ and 0 otherwise. The total cost for a traveler selecting a path $a \in \mathcal{A}^i$ will then be $c^{ia}(q) = \sum_{e \in \mathcal{E}} t^e(x^e) \cdot \delta^{eia}$.

---

[2]Throughout this paper, we use the term "reward" to describe the scenario where the agents aim to maximize $u^i$, and use "cost" when the agents aim to do the opposite.

**Incentive Design.** Despite the difference, we see that an equilibrium of both atomic and nonatomic games can be formulated as a solution to a corresponding VI problem in the following form

$$\sum_{i \in \mathcal{N}} \lambda^i \cdot \langle v^i(x_*), x^i - x_*^i \rangle \le 0, \quad \text{for all } x \in \mathcal{X} = \prod_{i \in \mathcal{N}} \mathcal{X}^i, \tag{3.3}$$

where $v^i$ and $\mathcal{X}^i$ denote different terms in the two types of games. Suppose that there exists an incentive designer aiming to induce a desired equilibrium. To this end, the designer can add incentives $\theta \in \Theta \subseteq \mathbb{R}^d$, which is assumed to enter the reward/cost functions and thus leads to a parameterized $v_\theta^i(x)$. We assume that the designer's objective is determined by a function $f : \Theta \times \mathcal{X} \to \mathbb{R}$. Denote $S(\theta)$ as the solution set to (3.3). We then obtain the uniform formulation of the incentive design problem for both atomic games and nonatomic games

$$\min_{\theta \in \Theta} f_*(\theta) = f(\theta, x_*), \quad \text{s.t.} \quad x_* \in S(\theta). \tag{3.4}$$

If the equilibrium problem admits multiple solutions, the agents may converge to different ones and it would be difficult to determine which one can better predict the behaviors of the agents without additional information. In this paper, we *first* consider the case where the game admits a unique equilibrium. Sufficient conditions under which the game admits a unique equilibrium will also be provided later. We would consider the non-unique case later and show that our algorithms can still become applicable by adding an appropriate regularizer in the cost function.

**Stochastic Environment.** In the aforementioned settings, $v_\theta^i(x)$ is a deterministic function. Although most MPEC algorithms in the optimization literature follow this deterministic setting, in this paper, we hope our algorithm can handle more realistic scenarios. Specifically, in the real world, the environment could be stochastic if some environment parameters that fluctuate over days. In a traffic system, for example, both worse weather and special events may affect the road condition, hence the travel time $v_\theta^i(x)$ experienced by the drivers. We expect our algorithm can still work in the face of such stochasticity. To this end, we assume that $v_\theta^i(x)$ represents the expected value of the cost function. On each day, however, the agents can only receive a noised feedback $\widehat{v}_\theta^i$ as an estimation. In the next section, we develop algorithms based on such noisy feedback.

## 4 Algorithm

We propose to update $\theta$ and $x$ simultaneously to improve the computational efficiency. The game dynamics at the lower level is modeled using the mirror descent method. Specifically, at the stage $k$, given $\theta_k$ and $x_k$, the agent first receives $v_{\theta_k}^i(x_k)$ as the feedback. After receiving the feedback, the agents update their strategies via

$$x_{k+1}^i = \operatorname*{argmax}_{x^i \in \mathcal{X}^i} \big\{ \langle v_{\theta_k}^i(x_k), x^i \rangle - 1/\beta_k^i \cdot D_{\psi^i}(x_k^i, x^i) \big\}, \tag{4.1}$$

where $D_{\psi^i}(x_k^i, x^i)$ is the Bregman divergence induced by a strongly convex function $\psi^i$. The accurate value of $\nabla f_*(\theta_k)$, the gradient of its objective function, equals

$$\nabla_\theta f(\theta_k, x_*(\theta_k)) + \big[\nabla_\theta x_*(\theta_k)\big]^\top \cdot \nabla_x f(\theta_k, x_*(\theta_k)),$$

which requires the exact lower-level equilibrium $x_*(\theta_k)$. However, at the stage $k$, we only have the current strategy $x_k$. Therefore, we also have to establish an estimator of $\nabla x_*(\theta_k)$ and $\nabla f_*(\theta_k)$ using $x_k$, the form of which will be specified later.

*Remark* 4.1. The standard gradient descent method is double-loop because at each $\theta_k$ it involves an inner loop for solving the exact value of $x_*(\theta_k)$ and then calculating the exact gradient.

### 4.1 Unconstrained Game

We first consider unconstrained games with $\mathcal{X}^i = \mathbb{R}^{d^i}$, for all $i \in \mathcal{N}$. We select $\psi^i(\cdot)$ as smooth function, i.e., there exists a constant $H_\psi \ge 1$ such that for all $i \in \mathcal{N}$ and $x^i, x^{i\prime} \in \mathcal{X}^i$,

$$\big\| \nabla \psi^i(x^i) - \nabla \psi^i(x^{i\prime}) \big\|_2 \le H_\psi \cdot \| x^i - x^{i\prime} \|_2. \tag{4.2}$$

Example of $\psi^i$ satisfying this assumption include (but is not limited to) $\psi^i(x^i) = (x^i)^\top Q^i x^i / 2$, where $Q^i \in \mathbb{R}^{d^i} \times \mathbb{R}^{d^i}$ is a positive definite matrix. It can be directly checked that we can set

$H_\psi = \max_{i \in \mathcal{N}} \delta^i$, where $\delta^i$ is the largest singular value of $Q^i$. In this case, the corresponding Bregman divergence becomes $D_{\psi^i}(x^i, x^{i'}) = (x^i - x^{i'})^\top Q^i (x^i - x^{i'})/2$, which is known as the squared Mahalanobis distance. Before laying out the algorithm, we first give the following lemma characterizing $\nabla_\theta x_*(\theta)$.

**Lemma 4.2.** *When $\mathcal{X}^i = \mathbb{R}^{d^i}$ and $\nabla_x v_\theta(x_*(\theta))$ is non-singular, it holds that*

$$\nabla_\theta x_*(\theta) = -\Big[\nabla_x v_\theta\big(x_*(\theta)\big)\Big]^{-1} \cdot \nabla_\theta v_\theta\big(x_*(\theta)\big).$$

*Proof.* See Appendix B.2 for detailed proof. $\qquad\square$

For any given $\theta \in \Theta$ and $x \in \mathcal{X}$, we define

$$\widetilde{\nabla} f(\theta, x) = \nabla_\theta f(\theta, x) - \big[\nabla_\theta v_\theta(x)\big]^\top \cdot \big[\nabla_x v_\theta(x)\big]^{-1} \cdot \nabla_x f(\theta, x). \tag{4.3}$$

Although we cannot obtain the exact value of $\nabla f_*(\theta_k)$, we may use $\widetilde{\nabla} f(\theta_k, x_k)$ as a *surrogate* and update $\theta_k$ based on $\widehat{\nabla} f_*(\theta_k, x_k)$ instead. Now we are ready to present the following bilevel incentive design algorithm for unconstrained games (see Algorithm 1).

---

**Algorithm 1** Bilevel incentive design for unconstrained games

---

  **Input:** $\theta_0 \in \Theta$, $x_0 \in \mathcal{X} = \mathbb{R}^d$, where $d = \sum_{i \in \mathcal{N}} d^i$, sequence of step sizes $(\alpha_k, \{\beta_k^i\}_{i \in \mathcal{N}})$.
  **For** $k = 0, 1, \dots$ **do:**
    Update strategy profile

$$x_{k+1}^i = \underset{x^i \in \mathcal{X}^i}{\arg\max}\big\{\langle \widehat{v}_k^i, x^i \rangle - 1/\beta_k^i \cdot D_{\psi^i}(x_k^i, x^i)\big\}, \tag{4.4}$$

    for all $i \in \mathcal{N}$, where $\widehat{v}_k^i$ is an estimator of $v_{\theta_k}^i(x_k)$.
    Update incentive parameter

$$\theta_{k+1} = \underset{\theta \in \Theta}{\arg\max}\big\{\langle \widetilde{\nabla} f(\theta_k, x_{k+1}), \theta \rangle - 1/\alpha_k \cdot \|\theta - \theta_k\|_2^2\big\}.$$

  **EndFor**
  **Output:** Last iteration incentive parameter $\theta_{k+1}$ and strategy profile $x_{k+1}$.

---

In Algorithm 1, if $\theta_k$ and $x_k$ converge to fixed points $\bar{\theta}$ and $\bar{x}$, respectively, then $\bar{x} = x_*(\bar{\theta})$ is expected to be satisfied. Thus, we would also have $\widehat{\nabla} f(\bar{\theta}, \bar{x}) = \nabla f_*(\bar{\theta})$. Thus, the optimality of $\bar{\theta}$ can be then guaranteed. It implies that the algorithm would find the optimal solution if it converges. Instead, the difficult part is how to design appropriate step sizes that ensure convergence. In this paper, we provide such conditions in Section 5.1.

### 4.2 Simplex-Constrained Game

We then consider the case where for all $i \in \mathcal{N}$, $x^i$ is constrained within the probability simplex

$$\Delta\big([d^i]\big) = \big\{x^i \in \mathbb{R}^{d^i} \,\big|\, x^i \geq 0, (x^i)^\top \mathbf{1}_{d^i} = 1\big\},$$

where $\mathbf{1}_{d^i} \in \mathbb{R}^{d^i}$ is the vector of all ones. Here we remark that any classic game-theoretic models are simplex-constrained. In fact, as long as the agent faces a finite number of choices and adopts a mixed strategy, its decision space would be a probability simplex [39]. In addition, some other types of decisions may also be constrained by a simplex. For example, financial investment concerns how to split the money on different assets. In such a scenario, the budget constraint can also be represented by a probability simplex.

In such a case, we naturally consider $\psi^i(x^i) = -\sum_{j \in [d^i]} [x^i]_j \cdot \log[x^i]_j$, which is the Shannon entropy. Such a choice gives the Bregman divergence $D_{\psi^i}(x^i, x^{i'}) = \sum_{j \in [d^i]} [x^i]_j \cdot \log([x^{i'}]_j / [x^i]_j)$, which is known as the KL divergence. In this case, we still first need to characterize $\nabla_\theta x_*(\theta)$, which also has an analytic form. Specifically, if we define a function $h_\theta(x) = (h_\theta^i(x^i))_{i \in \mathcal{N}}$ that satisfies

$$h_\theta^i(x^i) = \underset{x'^i \in \mathcal{X}^i}{\arg\max}\Big\{\big\langle v_\theta^i(x^i), x'^i \big\rangle - 1/\beta_k^i \cdot D_{\psi^i}(x^i, x'^i)\Big\},$$

then for any $\theta$, $x_*^i(\theta)$ satisfies $x_*^i(\theta) = h_\theta(x_*^i(\theta))$ [12]. Implicitly differentiating through this fixed point equation then yields $\nabla x_*(\theta) = \nabla_\theta h_\theta(x_*(\theta)) \cdot (I - \nabla_x h_\theta(x_*(\theta)))^{-1}$. Then, similar to (4.3), we may use

$$\widetilde{\nabla} f(\theta, x) = \nabla_\theta f(\theta, x) - \nabla_\theta h_\theta(x) \cdot \left(I - \nabla_x h_\theta(x)\right)^{-1} \cdot \nabla_x f(\theta, x) \tag{4.5}$$

to approximate the actual gradient $\nabla f_*(\theta)$ and then update $\theta_k$ based on $\nabla f_*(\theta_k)$ instead.

*Remark* 4.3. The mapping $h_\theta(x)$ has an analytic expression, which reads

$$h_\theta^i(x) = x^i \cdot \exp\!\left(-\beta_k^i \cdot v_\theta^i(x)\right) \Big/ \left\|x^i \cdot \exp\!\left(-\beta_k^i \cdot v_\theta^i(x)\right)\right\|_1.$$

Hence, both $\nabla_x h_\theta(x)$ and $\nabla_\theta h_\theta(x)$ can also be calculated analytically.

In addition to a different gradient estimate, we also modify Algorithm 1 to keep the iterations $x_k$ from hitting the boundary at the early stage. The modification involves an additional step that mixes the strategy with a uniform strategy $\mathbf{1}_{d^i}/d^i$, i.e., imposing an additional step

$$\widetilde{x}_{k+1}^i = (1 - \nu_{k+1}) \cdot x_{k+1} + \nu_{k+1} \cdot \mathbf{1}_{d^i}/d^i$$

upon finishing the update (4.4), where $\nu_{k+1} \in (0,1)$ is a the mixing parameter, decreasing to 0 when $k \to \infty$. In the following, we give the formal presentation of the modified bilevel incentive design algorithm for simplex-constrained games (see Algorithm 2).

---

**Algorithm 2** Bilevel incentive design for simplex constrained games

---

**Input:** $\theta_0 \in \Theta$, $x_0 \in \mathcal{X}$, step sizes $(\alpha_k, \{\beta_k^i\}_{i \in \mathcal{N}})$, $k \geq 0$, and mixing parameters $\nu_k, k \geq 0$.
**For** $k = 0, 1, \dots$ **do:**
    Update strategy profile

$$x_{k+1}^i = \operatorname*{argmax}_{x^i \in \Delta([d^i])} \left\{\langle \widehat{v}_k^i, x^i \rangle - 1/\beta_k^i \cdot D_{\psi^i}(\widetilde{x}_k^i, x^i)\right\},$$

$$\widetilde{x}_{k+1}^i = (1 - \nu_k) \cdot x_{k+1} + \nu_k \cdot \mathbf{1}_{d^i}/d^i, \tag{4.6}$$

    for all $i \in \mathcal{N}$, where $\widehat{v}_k^i$ is an estimator of $v_{\theta_k}^i(\widetilde{x}_k)$.
    Update incentive parameter

$$\theta_{k+1} \leftarrow \operatorname*{argmax}_{\theta \in \Theta} \left\{\langle \widetilde{\nabla} f(\theta_k, \widetilde{x}_{k+1}), \theta \rangle - 1/\alpha_k \cdot \|\theta - \theta_k\|_2^2\right\}.$$

**EndFor**
**Output:** Last iteration incentive parameter $\theta_{k+1}$ and strategy profile $x_{k+1}$.

---

Similar to Algorithm 1, at the core of the convergence of Algorithm 2 is still the step size. This case is even more complicated, as we need to design $\alpha_k$, $\beta_k$, and $\nu_k$ at the same time. In this paper, a provably convergent scheme is provided in Section 5.2.

Before closing this section, we remark that the algorithm can be easily adapted to other types of constraints by using another $h_\theta(x)$ to model the game dynamics. Particularly, the projected gradient descent dynamics has very broad applicability. In this case, the algorithm for calculating $\nabla_\theta h_\theta(x)$ and $\nabla_x h_\theta(x)$ is given by, for example, Amos and Kolter [2]. The additional step (4.6) then becomes unnecessary as it is dedicated to simplex constraints.

## 5 Convergence Analysis

In this section, we study the convergence of the proposed algorithms. For simplicity, define $\overline{D}_\psi(x, x') = \sum_{i \in \mathcal{N}} D_{\psi^i}(x^i, x^{i\prime})$. We make the following assumptions.

**Assumption 5.1.** The lower-level problem in (3.4) satisfies the following conditions. (1) The strategy set $\mathcal{X}^i$ of agent $i$ is a nonempty, compact, and convex subset of $\mathbb{R}^{d^i}$. (2) For each $i \in \mathcal{N}$, the gradient $v_\theta^i(\cdot)$ is $H_u$-Lipschitz continuous with respect to $\overline{D}_\psi$, i.e., for all $i \in \mathcal{N}$ and $x, x' \in \mathcal{X}$, $\|v_\theta^i(x) - v_\theta^i(x')\|_*^2 \leq H_u^2 \cdot \overline{D}_\psi(x, x')$. (3) There exist constants $\rho_\theta, \rho_x > 0$ such that for all $x \in \mathcal{X}$ and

$\theta \in \Theta$, $\|\nabla_\theta v_\theta(x)\|_2 < \rho_\theta$, and $\|[\nabla_x v_\theta(x)]^{-1}\|_2 \leq 1/\rho_x$. (4) For all $\theta \in \Theta$, the equilibrium $x_*(\theta)$ of the game is strongly stable with respect to $\overline{D}_\psi$, i.e., for all $x \in \mathcal{X}$, $\sum_{i \in \mathcal{N}} \lambda^i \cdot \langle v_\theta^i(x), x_*^i(\theta) - x^i \rangle \geq \overline{D}_\psi(x_*(\theta), x)$.

**Assumption 5.2.** The upper-level problem in (3.4) satisfies the following properties. (1) The set $\Theta$ is compact and convex. The function $f_*(\theta)$ is $\mu$-strongly convex and $\nabla f_*(\theta)$ has 2-norm uniformly bounded by $M$. (2) The extended gradient $\widetilde{\nabla} f(\theta, x)$ is $\widetilde{H}$-Lipschitz continuous with respect to $\overline{D}_\psi$, i.e., for all $x, x' \in \mathcal{X}$, $\|\widetilde{\nabla} f(\theta, x) - \widetilde{\nabla} f(\theta, x')\|_2^2 \leq \widetilde{H}^2 \cdot \overline{D}_\psi(x, x')$.

**Assumption 5.3.** Define the filtration by $\mathcal{F}_0^\theta = \{\theta_0\}$, $\mathcal{F}_0^x = \emptyset$, $\mathcal{F}_k^\theta = \mathcal{F}_{k-1}^\theta \cup \{x_{k-1}, \theta_k\}$, and $\mathcal{F}_k^x = \mathcal{F}_{k-1}^x \cup \{\theta_k, x_k\}$. We assume (1) the feedback $\widehat{v}_k$ is an unbiased estimate, i.e., for all $i \in \mathcal{N}$, we have $\mathbb{E}[\widehat{v}_k^i \mid \mathcal{F}_k^x] = v_{\theta_k}^i(x_k)$; (2) The feedback $\widehat{v}_k$ has bounded mean squared estimation errors, i.e., there exists $\delta_u > 0$ such that $\mathbb{E}[\|\widehat{v}_k^i - v_{\theta_k}^i(x_k)\|_*^2 \mid \mathcal{F}_k^x] \leq \delta_u^2$ for all $i \in \mathcal{N}$.

Below we discuss when the proposed assumptions hold, and if they are violated, how would our algorithm works. Assumption 5.1 includes the condition that $x_*(\theta)$ is strongly stable. In this case, it is the unique Nash equilibrium of the game [34]. It is also a common assumption in the analysis of the mirror descent dynamics itself [13]. We provide sufficient conditions for checking strong stability in Appendix B.1. We refer the readers to Section 6 for an explanation of how to extend our algorithm when this assumption is violated. Assumption 5.2 includes the convexity of the upper-level problem, which is usually a necessary condition to ensure global convergence. Yet, without convexity, our algorithm can still converge to a local minimum. Assumption 5.3 becomes unnecessary if we simply assume that the environment is deterministic. In this case, the accurate value of $v_\theta^i(x)$ is available. Yet, if the noises are added to the feedback, assuming that the noisy feedback is unbiased and bounded is still reasonable.

## 5.1 Unconstrained Game

In this part, we establish the convergence guarantee of Algorithm 1 for unconstrained games. We define the optimality gap $\epsilon_k^\theta$ and the equilibrium gap $\epsilon_{k+1}^x$ as

$$\epsilon_k^\theta := \mathbb{E}\big[\|\theta_k - \theta_*\|_2^2\big], \quad \epsilon_{k+1}^x := \mathbb{E}\big[\overline{D}_\psi\big(x_*(\theta_k), x_{k+1}\big)\big].$$

We track such two gaps as the convergence criteria in the subsequent results.

**Theorem 5.4.** *For Algorithm 1, set the step sizes* $\alpha_k = \alpha/(k+1)$, $\beta_k = \beta/(k+1)^{2/3}$, *and* $\beta_k^i = \lambda^i \cdot \beta_k$ *with constants* $\alpha > 0$ *and* $\beta > 0$ *satisfying*

$$\beta \leq 1/N \cdot H_u^2 \|\lambda\|_2^2, \quad \alpha/\beta^{3/2} \leq 1/12 \cdot H_\psi \widetilde{H} H_*,$$

*where* $H_* = \rho_\theta/\rho_x$. *Suppose that Assumptions 5.1-5.3 hold, then we have*

$$\epsilon_k^\theta = O(k^{-2/3}), \quad \epsilon_k^x = O(k^{-2/3}).$$

*Proof.* See Appendix C for detailed proof and a detailed expression of convergence rates. $\square$

## 5.2 Simplex-Constrained Game

In this part, we establish the convergence guarantee of Algorithm 2 for simplex constrained games. We still define optimality gap $\epsilon_k^\theta$ as $\epsilon_k^\theta = \mathbb{E}\big[\|\theta_k - \theta_*\|_2^2\big]$. Yet, corresponding to (4.6), we track $\widetilde{\epsilon}_{k+1}^x$ as a measure of convergence for the strategies of the agents, which is defined as

$$\widetilde{\epsilon}_{k+1}^x = \mathbb{E}\big[\overline{D}_\psi\big(\widetilde{x}_*(\theta_k), \widetilde{x}_{k+1}\big)\big],$$

where $\widetilde{x}_*(\theta_k) = (1 - \nu_k) \cdot x_*(\theta_k) + \nu_k \cdot \mathbf{1}/d^i$. We are now ready to give the convergence guarantee of Algorithm 2.

**Theorem 5.5.** *For Algorithm 2, set the step sizes* $\alpha_k = \alpha/(k+1)^{1/2}$, $\beta_k = \beta/(k+1)^{2/7}$, $\beta_k^i = \lambda^i \cdot \beta_k$, *and* $\nu_k = \nu/(k+1)^{4/7}$ *with constants* $\alpha > 0$ *and* $\beta > 0$ *satisfying*

$$\beta \leq 1/6 \cdot N H_u^2 \|\lambda\|_2^2, \quad \alpha/\beta^{3/2} \leq 1/7 \cdot \widetilde{H} \widetilde{H}_*,$$

where $\widetilde{H}_* = (1+d)\rho_\theta/\rho_x$. *Suppose that Assumptions 5.1-5.3 hold. If there exists some constant $V_* > 0$ such that $\|v_\theta(x_*(\theta))\|_\infty \leq V_*$ for any $\theta \in \Theta$, we then have*

$$\epsilon_k^\theta = O(k^{-2/7}), \quad \widetilde{\epsilon}_k^x = O(k^{-2/7}).$$

*Proof.* See Appendix D for detailed proof and a detailed form of the convergence rates. $\square$

## 6 Extensions to Games with Multiple Equilibria

We then briefly discuss how to apply our algorithms when the lower-level game has multiple equilibria.

**Case I**: If the function $v_\theta(x) = (v_\theta^i(x))_i$ is strongly monotone in the neighbourhood of each equilibrium, then all equilibria are strongly stable in a neighbourhood and hence isolated [38]. In this case, our algorithms can be directly applied as $\nabla_x v_\theta(x)$ is non-singular in these neighborhoods. It is commonly believed that the most likely equilibrium is the one reached by the game dynamics [52]; our algorithm naturally converges to this one.

**Case II**: If the function $v_\theta(x) = (v_\theta^i(x))_i$ is monotone but *not* strongly monotone, then the equilibrium set is a convex and closed region [34]. This case is challenging, as the matrix $\nabla_x v_\theta(x)$ needed to be inverted would become singular. Nevertheless, we can simply assume the agents are *bounded rational* [42, 1, 33]. The bounded rationality would result in a quantal response equilibrium for predicting the agents' response. We refer the readers to Appendix F for a detailed explanation (with numerical examples for illustration). Here we briefly sum up the key takeaways: (1) it is equivalent to add a regularizer $\eta^i \cdot (\log(x^i + \epsilon) + 1)$ to $v_\theta^i(x)$ for some $\eta^i > 0$ and $\epsilon \geq 0$; (2) as long as $\theta > 0$, the strong stability condition in Assumption 5.1 would then be satisfied, hence a unique equilibrium would exist; (3) as long as $\epsilon > 0$, the Lipschitz continuous condition in Assumption 5.1 will also not be violated. In a nutshell, the bounded rationality assumption can simultaneously make our model more realistic and satisfy the assumptions in Section 5.

## 7 Numerical Experiments

In this section, we conduct two numerical experiments to test our algorithms. All numerical results reported in this section were produced either on a MacBook Pro (15-inch, 2017) with 2.9 GHz Quad-Core Intel Core i7 CPU.

**Pollution Control via Emission Tax.** We first consider the oligopoly model introduced in Example 3.1. We assume that producing $a_i$ units of output, firm $i$ would generate $e_i = d^i a_i$ units of emissions. We consider the following social welfare function [44]

$$W(a) = \int_0^{\sum_{i=1}^n a^i} (p_0 - \gamma \cdot q)\, \mathrm{d}q - \sum_{i=1}^n c^i \cdot q^i - \tau \cdot \sum_{i=1}^n d^i a^i,$$

where the first term is the consumers' surplus, the second term is the total production cost, and the third term is the social damage caused by pollution. To maximize social welfare, an authority can impose emission taxes on the productions. Specifically, whenever producing $a_i$ units of output, firm $i$ could be charged $\pi^i \cdot a^i$, where $\pi^i$ is specialized for the firm $i$. In the experiment, we set $n = 100$, $p_0 = 100$, $\gamma = 1$, $\tau = 10$, and $d^i = 10 - \exp(c^i) + \epsilon^i$, where $\{c^i\}_{i=1}^{100}$ are evenly spaced between 1 and 2 and $\{\epsilon^i\}_{i=1}^{100}$ are the white noises. Under this setting, $c_i$ and $e_i$ are negatively correlated, which is realistic because if a firm hopes to reduce their pollution by updating their emission control systems, the production cost must be increased accordingly.

Through this small numerical example, we hope to illustrate that the single-loop scheme developed in this paper is indeed much more efficient than previous double-loop algorithms. To this end, we compare our algorithm with two double-loop schemes proposed by Li et al. [25]. In both approaches, the lower-level equilibrium problem is solved exactly first at each iteration. But afterwards, the upper-level gradients are respectively obtained via automatic differentiation (AD) and implicit differentiation (ID). To make a fair comparison, the *same* hyperparameters — including the initial solutions, the learning rates, and the tolerance values for both upper- and lower-level problems — are employed for the tested algorithms (double-loop AD, double-loop ID, and our algorithm).

Table 1 reports statistics related to the computation performance, including the total CPU time, the total iteration number, and the CPU time per iteration. The results reveal that all tested algorithm take a similar number of iterations to reach the same level of precision. However, the running time per iteration required by our algorithm is significantly lower than the two double-loop approaches. Hence, in general, our scheme is more efficient.

Table 1: Performance of the tested algorithms for solving the emission tax design problem.

| Method | Double-loop AD | Double-loop ID | Our algorithm |
|---|---|---|---|
| Total CPU time (s) | 71.56 | 11.29 | 2.09 |
| Total iteration number | 519 | 626 | 813 |
| Time per iteration (ms) | 137.89 | 18.04 | 2.57 |

**Second-Best Congestion Pricing**. We then consider the routing game model introduced in Example 3.2. To minimize the total travel delay, an authority on behalf of the public sector could impose appropriate tolls on selected roads [49]. This problem of determining tolls is commonly known as the congestion pricing problem. The second-best scheme assumes that only a subset of links can be charged [48]. Specifically, we write $\pi \in \mathbb{R}_+^{|\mathcal{E}|}$ as the toll imposed on all the links and $\mathcal{E}_{\text{toll}}$ as the set of tollable links. We model total cost for a traveler selecting a path $a \in \mathcal{A}$ as

$$c^{ia}(\pi, q) = \sum_{e \in \mathcal{E}} \big( t^e(x^e) + \pi^e \big) \cdot \delta^{eia} + \eta \cdot \big( \log(q^{ia}) + 1 \big),$$

where we add an extra term $(\log(q^{ia}) + 1)$ to characterize the uncertainties in travelers' route choices [20, 5]. It results in a quantal response equilibrium, as discussed in Section 6. We test our algorithm on a real-world traffic network: the Sioux-Falls network (See Lawphongpanich and Hearn [24] for its structure). We select 20 links (11, 35, 32, 68, 46, 21, 65, 52, 71, 74, 33, 64, 69, 14, 18, 39, 57, 48, 15, 51) for imposing congestion tolls.

We run Algorithm 2 to solve the problem and compare 4 different settings on the selection of step sizes. Setting A: $\alpha_k = \alpha/(k+1)^{1/2}$, $\beta_k = \beta/(k+1)^{2/7}$, and $\nu_k = \nu/(k+1)^{4/7}$. It ensures convergence according to Theorem 5.5. Setting B: $\alpha_k = \alpha/(k+1)^{1/2}$, $\beta_k = \beta/(k+1)^1$, and $\nu_k = \nu/(k+1)^{4/7}$. It only increases the decreasing rate of the step size in the inner loop. Setting C: $\alpha_k = \alpha/(k+1)^1$, $\beta_k = \beta/(k+1)^1$, and $\nu_k = \nu/(k+1)^1$. It assumes that all step sizes decrease based on the classic $O(1/k)$ rate. Setting D: $\alpha_k = \alpha/(k+1)^{1/2}$, $\beta_k = \beta/(k+1)^{2/7}$, and $\nu_k = 0$. It does not adopt the mixing step proposed in our paper. We add white noise to the costs received by the agents based on Gaussian distributions and run our algorithm under each setting with 10 times. The mean value of upper-level optimality gaps and lower-level equilibrium gaps are reported in Figure 1 (the shadowed areas are plotted based on "mean ± std" area over all sampled trajectories).

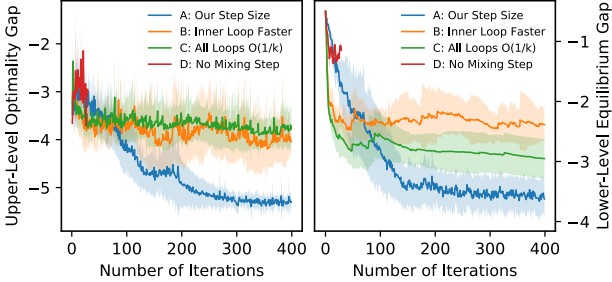

Figure 1: Comparison across different algorithms and step sizes when solving the second-best congestion pricing problem.

Below we summarize a few observations. First, without the mixing step proposed in our paper, the algorithm does hit the boundary too early. The algorithm fails after just a few iterations. It verifies our earlier claim that directly extending previous methods [21, 9] designed for bilevel optimization problems to MPECs is problematic. Second, the step size given by Theorem 5.5 ensures the fastest and the most stable convergence.

## Acknowledgments and Disclosure of Funding

Mingyi Hong's research is funded by NSF under the award numbers CIF-1910385 and CMMI-1727757. Yu (Marco) Nie's research is funded by NSF under the award number CMMI-2225087. Zhaoran Wang's research is funded by NSF under the award number ECCS-2048075.

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
