# A    Notations

We collect the most frequently used notations here.

| Category | Symbol | Definition |
|---|---|---|
| | $\mathcal{N}$ | set of all $N$ agents |
| | $x^i \in \mathcal{X}^i \subseteq \mathbb{R}^{d^i}$ | agent $i$'s strategy |
| Game | $u^i : \mathcal{X}^i \to \mathbb{R}$ | reward function of agent $i$ |
| (lower level) | $v^i : \mathcal{X}^i \to \mathbb{R}^{d^i}$ | $\nabla_{x^i} u^i$, unit reward of agent $i$ |
| | $\lambda^i \in \mathbb{R}_+$ | variational stability coefficient of agent $i$ |
| | $u_\theta^i, v_\theta^i$ | incentivized reward and unit reward |
| | $\theta \in \Theta$ | incentive distributed to the agents |
| Incentive designer | $f : \mathcal{X} \times \Theta \to \mathbb{R}$ | objective of the incentive designer |
| (upper level) | $x_* \in S(\theta)$ | equilibrium induced by incentive $\theta$ |
| | $f_* : \Theta \to \mathbb{R}$ | designer's objective value under $x_* \in S(\theta)$ |
| | $D_{\psi^i}$ | Bregman divergence in agent $i$'s strategy update |
| | $\beta_k^i$ | equilibrium learning rate for agent $i$ |
| Algorithm | $\alpha_k$ | incentive learning rate for the designer |
| ($k$-th iteration) | $\widehat{v}_k^i$ | estimate of incentivized unit cost $v_{\theta_k}^i(x_k^i)$ |
| | $\widetilde{\nabla} f$ | approximate incentive gradient |
| | $h_{\theta_k}^i$ | equilibrium learning dynamic |
| | $\widetilde{x}_{k+1}^i$ | agent $i$'s strategy mixed with uniform strategy |
| | $H_u$ | Lipschitz continuity coefficient of $v_\theta^i$ |
| | $\rho_\theta$ | spectral upper bound of $\nabla_\theta v_\theta$ |
| Analysis | $\rho_x$ | spectral lower bound of $\nabla_x v_\theta$ |
| (assumptions) | $\mu$ | strong convexity coefficient of $f_*$ |
| | $M$ | $\ell_2$ upper bound of $\nabla f_*$ |
| | $\widetilde{H}$ | Lipschitz continuity coefficient of $\widetilde{\nabla} f$ |
| | $\delta_u$ | estimation MSE upper bound of $\widehat{v}_k^i$ |

# B    Detailed Discussions on the Games

In this section, we present detailed discussions on the properties of the games.

## B.1    Strong Stability

We establish the following sufficient condition for strong stability.

**Lemma B.1.** *Define the matrix $H^\lambda(x)$ as a block matrix with $(i,j)$-th block taking the form of*

$$H_{i,j}^\lambda(x) = \lambda^i \cdot \nabla_{x^j} v^i(x).$$

*Suppose that $\psi^i$ satisfies the smooth condition (4.2). If $H^\lambda(x) + H^\lambda(x)^\top \prec -2 \cdot H_\psi \cdot I_d$ for some $\lambda \in \mathbb{R}_+^N$, then the Nash equilibrium $x_*$ is $\lambda$-strongly stable with respect to the Bregman divergence $\overline{D}_\psi$.*

*Proof.* We define the $\lambda$-weighted gradient of the game as

$$v^\lambda(x) = \mathrm{Diag}(\lambda^1 \cdot I_{d^1}, \ldots, \lambda^N \cdot I_{d^N}) \cdot v(x).$$

By definition, we can verify that $H^\lambda(x)$ is the Jacobian matrix of $v^\lambda(x)$ with respect to $x$. For any $x, x' \in \mathcal{X}$, let $x_\omega = \omega \cdot x + (1 - \omega) \cdot x'$. We have

$$v^\lambda(x) - v^\lambda(x') = \int_0^1 \frac{\mathrm{d}v^\lambda(x_\omega)}{\mathrm{d}\omega} \cdot \mathrm{d}\omega$$

$$= \int_0^1 H^\lambda(x_\omega) \cdot \frac{\mathrm{d}x_\omega}{\mathrm{d}\omega} \cdot \mathrm{d}\omega = \int_0^1 H^\lambda(x_\omega)(x - x') \cdot \mathrm{d}\omega. \qquad \text{(B.1)}$$

Taking additional dot product $\langle \cdot, x - x' \rangle$ on both sides of (B.1), we have

$$\langle v^\lambda(x) - v^\lambda(x'), x - x' \rangle = 1/2 \cdot \int_0^1 (x - x')^\top \big( H^\lambda(x_\omega) + H^\lambda(x_\omega)^\top \big)(x - x') \cdot \mathrm{d}\omega \tag{B.2}$$

$$\leq -\int_0^1 H_\psi \cdot \|x - x'\|_2^2 \cdot \mathrm{d}\omega = -H_\psi \cdot \|x - x'\|_2^2,$$

where the inequality follows from $H^\lambda(x_\omega) + H^\lambda(x_\omega)^\top \preceq -2 \cdot I_d$. Setting $x' = x_*$, we further have

$$\langle v^\lambda(x), x - x_* \rangle \leq \langle v^\lambda(x_*), x - x_* \rangle - H_\psi \cdot \|x - x_*\|_2^2 \leq -H_\psi \cdot \|x - x_*\|_2^2, \tag{B.3}$$

where the second inequality follows from the fact that $x_*$ is a Nash equilibrium. Eventually, as $\psi^i$ satisfies (4.2), we have

$$\overline{D}_\psi(x_*, x) = \sum_{i \in \mathcal{N}} D_{\psi_i}(x_*^i, x^i) \leq H_\psi \cdot \sum_{i \in \mathcal{N}} \|x^i - x_*^i\|_2^2 = H_\psi \cdot \|x - x_*\|_2^2. \tag{B.4}$$

Combining (B.3) and (B.4), we then conclude the proof. □

**Lemma B.2.** *Define the matrix $\widetilde{H}^\lambda(x)$ as*

$$\widetilde{H}^\lambda_{i,j}(x) = \lambda^i \cdot \nabla_{x^j} \widetilde{v}^i(x), \quad \text{where } \widetilde{v}^i(x) = v^i(x) + 1/\lambda^i \cdot \log x^i.$$

*Suppose that $\psi^i$ is the negative entropy function. If $\widetilde{H}^\lambda(x) + \widetilde{H}^\lambda(x)^\top \prec 0$ for some $\lambda \in \mathbb{R}_+^N$, then Nash equilibrium $x_*$ is $\lambda$-strongly stable with respect to the KL divergence.*

*Proof.* For any $x, x' \in \mathcal{X}$, let $x_\omega = \omega \cdot x + (1 - \omega) \cdot x'$. Similar to (B.2), we first obtain

$$\langle \widetilde{v}^\lambda(x) - \widetilde{v}^\lambda(x'), x - x' \rangle = 1/2 \cdot \int_0^1 (x - x')^\top \big( \widetilde{H}^\lambda(x_\omega) + \widetilde{H}^\lambda(x_\omega)^\top \big)(x - x') \cdot \mathrm{d}\omega \leq 0, \tag{B.5}$$

Noted that

$$\langle \log x^i - \log x^{i\prime}, x^i - x^{i\prime} \rangle = \langle \log x^i / x^{i\prime}, x^i \rangle + \langle \log x^{i\prime} / x^i, x^{i\prime} \rangle = \mathrm{KL}(x^i, x^{i\prime}) + \mathrm{KL}(x^{i\prime}, x^i),$$

by setting $x' = x_*$ in (B.5), we further have

$$\langle v^\lambda(x), x - x_* \rangle \leq \langle v^\lambda(x_*), x - x_* \rangle - \sum_{i=1}^n \mathrm{KL}(x_*^i, x^i) - \sum_{i=1}^n \mathrm{KL}(x^i, x_*^i) \leq -\sum_{i=1}^n \mathrm{KL}(x_*^i, x^i),$$

where the second inequality follows from the fact that $x_*$ is a Nash equilibrium. □

### B.2 Sensitivity of Nash Equilibrium

**Unconstrained vame.** We now provide the proof of Lemma 4.2.

*Proof of Lemma 4.2.* Since $\mathcal{X}^i = \mathbb{R}^{d^i}$ for all $i \in \mathcal{N}$, by the definition of $x_*(\theta)$, we have $v_\theta(x_*(\theta)) = 0$ for all $\theta \in \mathbb{R}^d$. Then, differentiating this equality with respect to $\theta$ on both ends, for any $i \in \mathcal{N}$, we have

$$\nabla^2_{x, x^i} u^i_\theta \big(x_*(\theta)\big) \nabla_\theta x_*(\theta) + \nabla^2_{\theta, x^i} u^i_\theta \big(x_*(\theta)\big) = 0.$$

Thus, we have

$$\nabla_\theta x_*(\theta) = -\big[ \nabla_x v_\theta(x) \big]^{-1} \nabla_\theta v_\theta(x) \big|_{x = x_*(\theta)}.$$

□

As a consequence of Lemma 4.2, we have the following lemma addressing the sensitivity of Nash equilibrium with respect to the incentive parameter $\theta$.

**Lemma B.3.** *Under Assumption 5.1, we have for $\{\widetilde{x}_k\}_{k \geq 0}$ generated by Algorithm 1,*

$$\big\| x_*(\theta_k) - x_*(\theta_{k-1}) \big\|_2 \leq H_* \cdot \|\theta_k - \theta_{k-1}\|_2,$$

*where $H_* = \rho_\theta / \rho_x$.*

*Proof.* We have for some $\overline{\theta} = \omega \cdot \theta_k + (1 - \omega) \cdot \theta_{k-1}$ with $\omega \in [0, 1]$,

$$
\begin{aligned}
\left\| x_*(\theta_k) - x_*(\theta_{k-1}) \right\|_2 &= \left\| \nabla_\theta x_*(\overline{\theta}) \right\|_2 \cdot \|\theta_k - \theta_{k-1}\|_2 \\
&= \left\| \left[ \nabla_x v_\theta \left( x_*(\overline{\theta}) \right) \right]^{-1} \nabla_\theta v_\theta \left( x_*(\overline{\theta}) \right) \right\|_2 \cdot \|\theta_k - \theta_{k-1}\|_2 \\
&\leq \left\| \left[ \nabla_x v_\theta \left( x_*(\overline{\theta}) \right) \right]^{-1} \right\|_2 \cdot \left\| \nabla_\theta v_\theta \left( x_*(\overline{\theta}) \right) \right\|_2 \cdot \|\theta_k - \theta_{k-1}\|_2 \\
&\leq \rho_x / \rho_\theta \cdot \|\theta_k - \theta_{k-1}\|_2,
\end{aligned}
$$

where the last inequality follows from Assumption 5.1. $\qquad\square$

**Simplex-Constrained Game.** We first provide the following lemma.

**Lemma B.4** (Theorem 1 in [41]). *When $\mathcal{X}^i = \Delta([d^i])$ and $\nabla_x v_\theta(x_*(\theta))$ is non-singular, the Jacobian of $x_*(\theta)$ takes the form*

$$
\nabla_\theta x_*(\theta) = -J_\theta \cdot \nabla_\theta v_\theta(x) \big|_{x = x_*(\theta)},
$$

*where*

$$
J_\theta = L - L A^\top [A L A^\top]^{-1} A L, \quad L = \left[ \nabla_x v_\theta \left( x_*(\theta) \right) \right]^{-1}.
$$

*Here $A = [\mathrm{blkdiag}(B^1, \ldots, B^n); \mathrm{blkdiag}(\mathbf{1}_{d^1}, \ldots, \mathbf{1}_{d^n})]$, where $B^i$ is obtained by deleting the rows $j$ in the identity matrix $I_{d^i}$ with $[x_*^i(\theta)]_j = 0$.*

As a consequence of Lemma B.4, we have the following lemma as the simplex constrained version of Lemma B.3.

**Lemma B.5.** *Under Assumption 5.1, we have*

$$
\left\| x_*(\theta_k) - x_*(\theta_{k-1}) \right\|_1 \leq \widetilde{H}_* \cdot \|\theta_k - \theta_{k-1}\|_2,
$$

*where $\widetilde{H}_* = (1 + d)\rho_\theta / \rho_x$.*

*Proof.* Recall that $J_\theta$ is defined in Lemma B.4 and that $\| \cdot \|_2$ is the spectral norm when operating on a matrix. We have

$$
\begin{aligned}
\|J_\theta\|_2 &\leq \|L\|_2 + \left\| L A^\top [A L A^\top]^{-1} A L \right\|_2 \\
&\leq \|L\|_2 + \|L\|_2 \cdot \mathrm{tr}\left( A^\top [A L A^\top]^{-1} A L \right) \\
&\leq \|L\|_2 + \|L\|_2 \cdot \mathrm{tr}\left( [A L A^\top]^{-1} A L A^\top \right) \\
&= (1 + d) \cdot \|L\|_2 \leq (1 + d)/\rho_x.
\end{aligned}
$$

Thus, by Lemma B.4, we have for some $\overline{\theta} = \omega \cdot \theta_k + (1 - \omega) \cdot \theta_{k-1}$ with $\omega \in [0, 1]$,

$$
\begin{aligned}
\|x_*(\theta_k) - x_*(\theta_{k-1})\| &\leq \left\| \nabla_\theta x_*(\overline{\theta}) \right\|_2 \cdot \|\theta_k - \theta_{k-1}\|_2 \\
&\leq \|J_\theta\|_2 \cdot \left\| \nabla_\theta v_\theta(x) \right\|_2 \cdot \|\theta_k - \theta_{k-1}\|_2 \\
&\leq (1 + d)\rho_\theta / \rho_x \cdot \|\theta_k - \theta_{k-1}\|_2,
\end{aligned}
$$

where the last inequality follows from Assumption 5.1. $\qquad\square$

# C  Proof of Theorem 5.4

We first present the following two lemmas under the conditions presented in Theorem 5.4.

**Lemma C.1.** *For all $k \geq 0$, we have*

$$
\epsilon_{k+1}^x \leq \epsilon_0^x \cdot \prod_{j=0}^{k} (1 - \beta_j/8) + \left[ M^2/4\widetilde{H}^2 + \delta_u^2 \|\lambda\|_2^2 \right] \cdot \sum_{l=0}^{k} \left[ \beta_l^2 \cdot \prod_{j=l+1}^{k} (1 - \beta_j/8) \right].
$$

*Proof.* See Appendix C.1 for detailed proof. □

**Lemma C.2.** *For all $k \geq 0$, we have*

$$\epsilon_{k+1}^\theta \leq \prod_{j=0}^k (1 - \mu\alpha_j) \cdot \epsilon_0^\theta + 2M^2 \cdot \sum_{l=0}^k \left[ \alpha_l^2 \cdot \prod_{j=l+1}^k (1 - \mu\alpha_j) \right]$$

$$+ \widetilde{H}^2 \cdot \sum_{l=0}^k \left[ (2\alpha_l^2 + \alpha_l/\mu) \cdot \epsilon_{l+1}^x \cdot \prod_{j=l+1}^k (1 - \mu\alpha_j) \right].$$

*Proof.* See Appendix C.2 for detailed proof. □

Now we are ready to present the proof of Theorem 5.4.

*Proof of Theorem 5.4.* For $\beta_k = \beta/(k+1)^{2/3}$, by [[21], Lemma C.4], we have

$$\prod_{j=1}^k (1 - \beta_j/8) \leq \sum_{l=0}^k \left[ \beta_l^2 \cdot \prod_{j=l+1}^k (1 - \beta_j/8) \right] \leq 8\beta_k.$$

Thus, by Lemma C.1, we have

$$\epsilon_{k+1}^x \leq \epsilon_0^x \cdot \prod_{j=0}^k (1 - \beta_j/8) + \left[ M^2/4\widetilde{H}^2 + \delta_u^2 \|\lambda\|_2^2 \right] \cdot \sum_{l=0}^k \left[ \beta_l^2 \cdot \prod_{j=l+1}^k (1 - \beta_j/4) \right]$$

$$\leq \left[ 8\epsilon_0^x + 2M^2/\widetilde{H}^2 + 8\delta_u^2 \|\lambda\|_2^2 \right] \cdot \beta_k = O(k^{-2/3}). \tag{C.1}$$

Similarly, we have for $\alpha_k = \alpha/(k+1)$,

$$\prod_{j=0}^k (1 - \mu\alpha_j) \leq \sum_{l=0}^k \left[ \alpha_l^2 \cdot \prod_{j=l+1}^k (1 - \mu\alpha_j) \right] \leq \sum_{l=0}^k \left[ \alpha_l^{5/3} \cdot \prod_{j=l+1}^k (1 - \mu\alpha_j) \right] \leq 2/\mu \cdot \alpha_k^{2/3}.$$

Thus, combining (C.1) with Lemma C.2, we obtain

$$\epsilon_{k+1}^\theta \leq \prod_{j=0}^k (1 - \mu\alpha_j) \cdot \epsilon_0^\theta + 2M^2 \cdot \sum_{l=0}^k \left[ \alpha_l^2 \cdot \prod_{j=l+1}^k (1 - \mu\alpha_j) \right] \tag{C.2}$$

$$+ \widetilde{H}^2 \cdot \sum_{l=0}^k \left[ (2\alpha_l^2 + \alpha_l/\mu) \cdot \epsilon_{l+1}^x \cdot \prod_{j=l+1}^k (1 - \mu\alpha_j) \right]$$

$$\leq 2\epsilon_0^\theta/\mu \cdot \alpha_k^{2/3} + (2M^2 + 2\widetilde{H}^2) \cdot \sum_{l=0}^k \left[ \alpha_l^2 \cdot \prod_{j=l+1}^k (1 - \mu\alpha_j) \right]$$

$$+ \left[ 8\epsilon_0^x + 2M^2/\widetilde{H}^2 + 8\delta_u^2 \|\lambda\|_2^2 \right] \cdot \widetilde{H}^2/\mu \cdot \sum_{l=0}^k \left[ \alpha_l \beta_{l+1} \cdot \prod_{j=l+1}^k (1 - \mu\alpha_j) \right]$$

$$\leq 2\epsilon_0^\theta/\mu \cdot \alpha_k^{2/3} + 4 \cdot (M^2 + \widetilde{H}^2)/\mu \cdot \alpha_k^{2/3}$$

$$+ \left[ 8\epsilon_0^x + 2M^2/\widetilde{H}^2 + 8\delta_u^2 \|\lambda\|_2^2 \right] \cdot 2\beta\widetilde{H}^2/\alpha^{2/3}\mu^2 \cdot \alpha_k^{2/3} = O(k^{-2/3}).$$

Here the third inequality follows from $\beta_k = \beta/\alpha^{2/3} \cdot \alpha_k^{2/3}$. Therefore, (C.1) and (C.2) conclude the proof of Theorem 5.4. □

## C.1   Proof of Lemma C.1

*Proof.* Recall that

$$x_{k+1}^i = \operatorname*{argmax}_{x^i \in \mathcal{X}^i} \left\{ \langle \widehat{v}_k^i, x^i \rangle - 1/\beta_k^i \cdot D_{\psi^i}(x_k^i, x^i) \right\}.$$

We have
$$D_{\psi^i}\big(x_*^i(\theta_k), x_{k+1}^i\big) \le D_{\psi^i}\big(x_*^i(\theta_k), x_k^i\big) - \beta_k^i \cdot \big\langle \widehat{v}_k^i, x_*^i(\theta_k) - x_{k+1}^i \big\rangle - D_{\psi^i}\big(x_k^i, x_{k+1}^i\big)$$
$$\le D_{\psi^i}\big(x_*^i(\theta_k), x_k^i\big) - \beta_k^i \cdot \big\langle \widehat{v}_k^i, x_*^i(\theta_k) - x_k^i \big\rangle$$
$$- \beta_k^i \cdot \big\langle \widehat{v}_k^i, x_k^i - x_{k+1}^i \big\rangle - 1/2 \cdot \|x_k^i - x_{k+1}^i\|^2,$$

where the second inequality follows from the fact that $D_{\psi^i}(x^i, x^{i\prime}) \ge 1/2 \cdot \|x^i - x^{i\prime}\|^2$. Taking the conditional expectation given $\mathcal{F}_k^x$, we obtain

$$\mathbb{E}\Big[D_{\psi^i}\big(x_*^i(\theta_k), x_{k+1}^i\big) \,\Big|\, \mathcal{F}_k^x\Big] \le D_{\psi^i}\big(x_*^i(\theta_k), x_k^i\big) - \beta_k^i \cdot \mathbb{E}\Big[\big\langle \widehat{v}_k^i, x_*^i(\theta_k) - x_k^i \big\rangle \,\Big|\, \mathcal{F}_k^x\Big] \tag{C.3}$$
$$+ \mathbb{E}\Big[-\beta_k^i \cdot \big\langle \widehat{v}_k^i, x_k^i - x_{k+1}^i \big\rangle - 1/2 \cdot \big\|x_k^i - x_{k+1}^i\big\|^2 \,\Big|\, \mathcal{F}_k^x\Big]$$
$$\le D_{\psi^i}\big(x_*^i(\theta_k), x_k^i\big) - \beta_k^i \cdot \big\langle v_{\theta_k}^i(x_k), x_*^i(\theta_k) - x_k^i \big\rangle + (\beta_k^i)^2/2 \cdot \mathbb{E}\big[\|\widehat{v}_k^i\|_*^2 \,\big|\, \mathcal{F}_k^x\big],$$

where the second inequality follows from Assumption 5.3. Summing up (C.3) for all $i \in \mathcal{N}$, we have

$$\mathbb{E}\Big[\overline{D}_\psi\big(x_*(\theta_k), x_{k+1}\big) \,\Big|\, \mathcal{F}_k^x\Big] \le \overline{D}_\psi\big(x_*(\theta_k), x_k\big) - \beta_k \cdot \sum_{i \in \mathcal{N}} \lambda^i \cdot \big\langle v_{\theta_k}^i(x_k), x_*^i(\theta_k) - x_k^i \big\rangle$$
$$+ \beta_k^2/2 \cdot \sum_{i \in \mathcal{N}} (\lambda^i)^2 \cdot \mathbb{E}\big[\|\widehat{v}_k^i\|_*^2 \,\big|\, \mathcal{F}_k^x\big]. \tag{C.4}$$

By the $\lambda$-strong variational stability of $x_i(\theta_k)$, we have

$$-\sum_{i \in \mathcal{N}} \lambda^i \cdot \big\langle v_{\theta_k}^i(x_k), x_*^i(\theta_k) - x_k^i \big\rangle \le -\overline{D}_\psi\big(x_*(\theta_k), x_k\big). \tag{C.5}$$

Moreover, by Assumption 5.3 we have

$$\mathbb{E}\big[\|\widehat{v}_k^i\|_*^2 \,\big|\, \mathcal{F}_k^x\big] \le 2 \cdot \mathbb{E}\Big[\big\|\widehat{v}_k^i - v_{\theta_k}^i(x_k)\big\|_*^2 + \big\|v_{\theta_k}^i(x_k)\big\|_*^2 \,\Big|\, \mathcal{F}_k^x\Big] \le 2\delta_u^2 + 2\big\|v_{\theta_k}^i(x_k)\big\|_*^2,$$

summing up which gives

$$\sum_{i \in \mathcal{N}} (\lambda^i)^2 \cdot \mathbb{E}\big[\|\widehat{v}_k^i\|_*^2 \,\big|\, \mathcal{F}_k^x\big] \le 2\delta_u^2 \cdot \sum_{i \in \mathcal{N}} (\lambda^i)^2 + 2 \cdot \sum_{i \in \mathcal{N}} (\lambda^i)^2 \cdot \Big\|v_{\theta_k}^i(x_k) - v_{\theta_k}^i\big(x_*(\theta_k)\big)\Big\|_*^2$$
$$\le 2\delta_u^2 \cdot \sum_{i \in \mathcal{N}} (\lambda^i)^2 + 2NH_u^2 \cdot \sum_{i \in \mathcal{N}} (\lambda^i)^2 \cdot \overline{D}_\psi\big(x_*(\theta_k), x_k\big)$$
$$= 2\delta_u^2 \|\lambda\|_2^2 + 2NH_u^2 \|\lambda\|_2^2 \cdot \overline{D}_\psi\big(x_*(\theta_k), x_k\big), \tag{C.6}$$

where the first inequality follows from the optimality condition that $v_{\theta_k}(x_*(\theta_k)) = 0$, and the second inequality follows from Assumption 5.1. Thus, taking (C.5) and (C.6) into (C.4), we obtain

$$\mathbb{E}\Big[\overline{D}_\psi\big(x_*(\theta_k), x_{k+1}\big) \,\Big|\, \mathcal{F}_k^x\Big] \tag{C.7}$$
$$\le (1 - \beta_k) \cdot \overline{D}_\psi\big(x_*(\theta_k), x_k\big) + \delta_u^2 \|\lambda\|_2^2 \beta_k^2 + 2NH_u^2 \|\lambda\|_2^2 \beta_k^2 \cdot \overline{D}_\psi\big(x_*(\theta_k), x_k\big)^2$$
$$\le \big(1 - \beta_k + NH_u^2 \|\lambda\|_2^2 \beta_k^2\big) \cdot \overline{D}_\psi\big(x_*(\theta_k), x_k\big) + \delta_u^2 \|\lambda\|_2^2 \beta_k^2$$
$$\le (1 - \beta_k/2) \cdot \overline{D}_\psi\big(x_*(\theta_k), x_k\big) + \delta_u^2 \|\lambda\|_2^2 \beta_k^2,$$

where the last inequality holds with $NH_u^2 \|\lambda\|_2^2 \beta_k^2 \le \beta_k$, which is satisfied by $\beta \le 1/NH_u^2 \|\lambda\|_2^2$. By Lemma E.1, we have for any $\gamma > \max\{1, H_\psi^2\}$,

$$\overline{D}_\psi\big(x_*(\theta_k), x_k\big) - \left(1 + \frac{1}{\gamma}\right) \cdot \overline{D}_\psi\big(x_*(\theta_{k-1}), x_k\big) \tag{C.8}$$
$$\le \frac{H_\psi^2 \cdot (1+\gamma)^2 - (1+\gamma)}{2\gamma} \cdot \sum_{i \in \mathcal{N}} \big\|x_*^i(\theta_k) - x_*^i(\theta_{k-1})\big\|^2$$
$$\le \frac{H_\psi^2 \cdot (1+\gamma)^2 - (1+\gamma)}{2\gamma} \cdot \big\|x_*(\theta_k) - x_*(\theta_{k-1})\big\|^2$$
$$\le \frac{H_\psi^2 \cdot (1+\gamma)^2 - (1+\gamma)}{2\gamma} \cdot H_*^2 \cdot \|\theta_k - \theta_{k-1}\|_2^2 \le \frac{H_\psi^2 \cdot (1+\gamma)^2 - (1+\gamma)}{2\gamma} \cdot H_*^2 \alpha_{k-1}^2 \cdot \|\widetilde{\nabla} f_{k-1}\|_2^2,$$

where the third inequality follows from Lemma B.3, and the last inequality follows from the fact that proximal mapping is non-expensive. Taking (C.8) into (C.7) and choosing $\gamma = (4 - 2\beta_k)/\beta_k$, we obtain

$$\mathbb{E}\Big[\overline{D}_\psi\big(x_*(\theta_k), x_{k+1}\big) \,\Big|\, \mathcal{F}_k^x\Big] \le (1 - \beta_k/2) \cdot \left(1 + \frac{1}{\gamma}\right) \cdot \overline{D}_\psi\big(x_*(\theta_{k-1}), x_k\big) + \delta_u^2 \|\lambda\|_2^2 \beta_k^2 \qquad \text{(C.9)}$$

$$+ (1 - \beta_k/2) \cdot \frac{H_\psi^2 \cdot (1+\gamma)^2 - (1+\gamma)}{2\gamma} \cdot H_*^2 \alpha_{k-1}^2 \cdot \mathbb{E}\big[\|\widetilde{\nabla} f_{k-1}\|_2^2 \,\big|\, \mathcal{F}_k^x\big],$$

$$\le (1 - \beta_k/4) \cdot \overline{D}_\psi\big(x_*(\theta_{k-1}), x_k\big) + \delta_u^2 \|\lambda\|_2^2 \beta_k^2$$

$$+ (1 - \beta_k/4) \cdot \frac{H_\psi^2 \cdot (1+\gamma) - 1}{2} \cdot H_*^2 \alpha_{k-1}^2 \cdot \mathbb{E}\big[\|\widetilde{\nabla} f_{k-1}\|_2^2 \,\big|\, \mathcal{F}_k^x\big].$$

By Assumptions 5.2 and 5.3, we have

$$\mathbb{E}\big[\|\widetilde{\nabla} f_{k-1}\|_2^2 \,\big|\, \mathcal{F}_{k-1}^\theta\big] \qquad\qquad\qquad \text{(C.10)}$$

$$\le 2\mathbb{E}\Big[\big\|\widetilde{\nabla} f(\theta_{k-1}, x_k) - \nabla f_*(\theta_{k-1})\big\|_2^2 + \big\|\nabla f_*(\theta_{k-1})\big\|_2^2 \,\Big|\, \mathcal{F}_{k-1}^\theta\Big]$$

$$\le 2\widetilde{H}^2 \cdot \overline{D}_\psi\big(x_*(\theta_{k-1}), x_k\big) + 2M^2,$$

taking which into (C.9) and taking expectation on both sides, we further obtain

$$\epsilon_{k+1}^x \le \Big\{1 - \beta_k/4 + \big[H_\psi^2 \cdot (1+\gamma) - 1\big] \cdot \widetilde{H}^2 H_*^2 \alpha_{k-1}^2\Big\} \cdot \epsilon_k^x + \delta_u^2 \|\lambda\|_2^2 \beta_k^2$$

$$+ (1 - \beta_k/4) \cdot \frac{H_\psi^2 \cdot (1+\gamma) - 1}{2} \cdot H_*^2 \alpha_{k-1}^2 \cdot 2M^2.$$

By $\gamma = (4 - 2\beta_k)/\beta_k$, we have

$$\big[H_\psi^2 \cdot (1+\gamma) - 1\big] \cdot \widetilde{H}^2 H_*^2 \alpha_{k-1}^2 = \big[H_\psi^2 \cdot (4/\beta_k - 1) - 1\big] \cdot \widetilde{H}^2 H_*^2 \alpha_{k-1}^2$$

$$\le 16 H_\psi^2 \widetilde{H}^2 H_*^2 \alpha_k^2/\beta_k \le \beta_k^2/8 \le \beta_k/8,$$

where the first inequality follows from $\alpha_{k-1} \le 2\alpha_k$, and the second inequality follows from $\alpha_k = \alpha/(k+1)$, $\beta_k = \beta/(k+1)^{2/3}$, and $\alpha/\beta^{3/2} \le 1/12 H_\psi \widetilde{H} H_*$. Thus, we obtain

$$\epsilon_{k+1}^x \le (1 - \beta_k/8) \cdot \epsilon_k^x + \big[M^2/4\widetilde{H}^2 + \delta_u^2 \|\lambda\|_2^2\big] \cdot \beta_k^2. \qquad \text{(C.11)}$$

Recursively applying (C.11), we obtain

$$\epsilon_{k+1}^x \le \epsilon_0^x \cdot \prod_{j=0}^k (1 - \beta_j/8) + \big[M^2/4\widetilde{H}^2 + \delta_u^2 \|\lambda\|_2^2\big] \cdot \sum_{l=0}^k \left[\beta_l^2 \cdot \prod_{j=l+1}^k (1 - \beta_j/4)\right].$$

Thus, we conclude the proof Lemma C.1. $\qquad\qquad\qquad\qquad\qquad\qquad \square$

## C.2 Proof of Lemma C.2

*Proof.* Since the projection $\arg\max_{x \in \mathcal{X}}$ is non-expansive, we have

$$\|\theta_{k+1} - \theta_*\|^2 \le \|\theta_k + \alpha_k \cdot \widehat{\nabla} f_k - \theta_*\|_2^2$$

$$= \|\theta_k - \theta_*\|_2^2 + \alpha_k^2 \cdot \|\widehat{\nabla} f_k\|_2^2 + 2\alpha_k \cdot \langle \widehat{\nabla} f_k, \theta_k - \theta_*\rangle.$$

Taking the conditional expectation given $\mathcal{F}_k^\theta$, we obtain

$$\mathbb{E}\big[\|\theta_{k+1} - \theta_*\|^2 \,\big|\, \mathcal{F}_k^\theta\big] \le \|\theta_k - \theta_*\|_2^2 + 2\alpha_k \cdot \big\langle \nabla_\theta f_*(\theta_k), \theta_k - \theta_*\big\rangle + \alpha_k^2 \cdot \mathbb{E}\big[\|\widetilde{\nabla} f_k\|_2^2 \,\big|\, \mathcal{F}_k^\theta\big]$$

$$+ 2\alpha_k \cdot \mathbb{E}\Big[\big\langle \widetilde{\nabla} f(\theta_k, x_{k+1}) - \nabla_\theta f_*(\theta_k), \theta_k - \theta_*\big\rangle \,\Big|\, \mathcal{F}_k^\theta\Big]$$

$$\le (1 - 2\mu\alpha_k) \cdot \|\theta_k - \theta_*\|_2^2 + \alpha_k^2 \cdot \mathbb{E}\big[\|\widetilde{\nabla} f_k\|_2^2 \,\big|\, \mathcal{F}_k^\theta\big]$$

$$+ \alpha_k/\mu \cdot \mathbb{E}\Big[\big\|\widetilde{\nabla} f(\theta_k, x_{k+1}) - \nabla_\theta f_*(\theta_k)\big\|_2 \,\Big|\, \mathcal{F}_k^\theta\Big] + \mu\alpha_k \cdot \|\theta_k - \theta_*\|_2^2$$

$$\le (1 - \mu\alpha_k) \cdot \|\theta_k - \theta_*\|_2^2 + \widetilde{H}^2 \alpha_k/\mu \cdot \mathbb{E}\Big[\overline{D}_\psi\big(x_*(\theta_k), x_{k+1}\big) \,\Big|\, \mathcal{F}_k^\theta\Big]$$

$$+ \alpha_k^2 \cdot \mathbb{E}\big[\|\widetilde{\nabla} f_k\|_2^2 \,\big|\, \mathcal{F}_k^\theta\big], \qquad\qquad\qquad \text{(C.12)}$$

where the second inequality follows from Assumption 5.2 and the Cauchy-Schwartz inequality, and the last inequality follows from Assumption 5.2. Applying (C.10) to (C.12) and taking expectation on both sides, we get

$$\epsilon_{k+1}^\theta \leq (1 - \mu\alpha_k) \cdot \epsilon_k^\theta + 2M^2 \cdot \alpha_k^2 + \widetilde{H}^2 \cdot (2\alpha_k^2 + \alpha_k/\mu) \cdot \epsilon_{k+1}^x. \tag{C.13}$$

Recursively applying (C.13), we obtain

$$\epsilon_{k+1}^\theta \leq \prod_{j=0}^k (1 - \mu\alpha_j) \cdot \epsilon_0^\theta + 2M^2 \cdot \sum_{l=0}^k \left[ \alpha_l^2 \cdot \prod_{j=l+1}^k (1 - \mu\alpha_j) \right]$$
$$+ \widetilde{H}^2 \cdot \sum_{l=0}^k \left[ (2\alpha_l^2 + \alpha_l/\mu) \cdot \epsilon_{l+1}^x \cdot \prod_{j=l+1}^k (1 - \mu\alpha_j) \right].$$

Thus, we conclude the proof of Lemma C.2. □

## D Proof of Theorem 5.5

We first present the following two lemmas under the conditions presented in Theorem 5.5

**Lemma D.1.** *For all $k \geq 0$, we have*

$$\widetilde{\epsilon}_{k+1}^x \leq \widetilde{\epsilon}_0^x \cdot \prod_{j=0}^k (1 - \beta_j/8) + \left[ (\delta_u^2 + 3V_*^2) \cdot \|\lambda\|_2^2 + (M^2 + 3N\widetilde{H}^2)/4\widetilde{H}^2 \right] \cdot \left[ \sum_{l=0}^k \beta_l^2 \cdot \prod_{j=l+1}^k (1 - \beta_j/8) \right]$$
$$+ N \cdot \left[ \sum_{l=0}^k \left( \beta_l \nu_l \log(1/\nu_l) + 2\nu_{l+1} + 2\nu_l^2 \right) \cdot \prod_{j=l+1}^k (1 - \beta_j/8) \right].$$

*Proof.* See Appendix D.1 for detailed proof. □

**Lemma D.2.** *For all $k \geq 0$, we have*

$$\epsilon_{k+1}^\theta \leq \prod_{j=0}^k (1 - \mu\alpha_j) \cdot \epsilon_0^\theta + (3M^2 + 6N\widetilde{H}^2) \cdot \sum_{l=0}^k \left[ \alpha_l^2 \cdot \prod_{j=l+1}^k (1 - \mu\alpha_j) \right]$$
$$+ \widetilde{H}^2 \cdot \sum_{l=0}^k \left[ (2\alpha_l^2 + \alpha_l/\mu) \cdot \epsilon_{l+1}^x \cdot \prod_{j=l+1}^k (1 - \mu\alpha_j) \right].$$

*Proof.* See Appendix D.2 for detailed proof. □

Now we are ready to prove Theorem 5.5.

*Proof of Theorem 5.5.* For $\beta_k = \beta/(k+1)^{2/7}$, by [[21], Lemma C.4], we have

$$\prod_{j=1}^k (1 - \beta_j/8) \leq \sum_{l=0}^k \left[ \beta_l^2 \cdot \prod_{j=l+1}^k (1 - \beta_j/8) \right] \leq 8\beta_k.$$

Thus, by Lemma C.1, we have

$$\widetilde{\epsilon}_{k+1}^x \leq \widetilde{\epsilon}_0^x \cdot \prod_{j=0}^k (1 - \beta_j/8) + \left[ (\delta_u^2 + 3V_*^2) \cdot \|\lambda\|_2^2 + (M^2 + 3N\widetilde{H}^2)/4\widetilde{H}^2 \right] \cdot \left[ \sum_{l=0}^k \beta_l^2 \cdot \prod_{j=l+1}^k (1 - \beta_j/8) \right]$$
$$+ N \cdot \left[ \sum_{l=0}^k \left( \beta_l \nu_l \log(1/\nu_l) + 2\nu_{l+1} + 2\nu_l^2 \right) \cdot \prod_{j=l+1}^k (1 - \beta_j/8) \right]$$
$$\leq \left[ 8\widetilde{\epsilon}_0^x + 8(\delta_u^2 + 3V_*^2) \cdot \|\lambda\|_2^2 + (2M^2 + 6N\widetilde{H}^2)/\widetilde{H}^2 \right] \cdot \beta_k$$
$$+ N \cdot \left[ \sum_{l=0}^k \left( \beta_l \nu_l \log(1/\nu_l) + 2\nu_{l+1} + 2\nu_l^2 \right) \cdot \prod_{j=l+1}^k (1 - \beta_j/8) \right]. \tag{D.1}$$

Since $\nu_l = 1/(l+1)^{4/7}$, we have

$$\beta_l \nu_l \log(1/\nu_l) + 2\nu_{l+1} + 2\nu_l^2 \leq (4/\beta^2 + 4/7\beta) \cdot \beta_l^2,$$

taking which into (D.1), we obtain

$$\widetilde{\epsilon}_{k+1}^x \leq \left[ 8\widetilde{\epsilon}_0^x + 8(\delta_u^2 + 3V_*^2) \cdot \|\lambda\|_2^2 + (2M^2 + 6N\widetilde{H}^2)/\widetilde{H}^2 \right] \cdot \beta_k + 32N \cdot (1/\beta^2 + 1/7\beta) \cdot \beta_k$$
$$= \widetilde{c} \cdot \beta_k = O(k^{-2/7}), \tag{D.2}$$

where

$$\widetilde{c} = 8\widetilde{\epsilon}_0^x + 8(\delta_u^2 + 3V_*^2) \cdot \|\lambda\|_2^2 + (2M^2 + 6N\widetilde{H}^2)/\widetilde{H}^2 + 32N \cdot (1/\beta^2 + 4/7\beta).$$

Similarly, we have for $\alpha_k = \alpha/(k+1)^{1/2}$,

$$\prod_{j=0}^{k}(1 - \mu\alpha_j) \leq \sum_{l=0}^{k}\left[\alpha_l^2 \cdot \prod_{j=l+1}^{k}(1 - \mu\alpha_j)\right] \leq \sum_{l=0}^{k}\left[\alpha_l^{11/7} \cdot \prod_{j=l+1}^{k}(1 - \mu\alpha_j)\right] \leq 2/\mu \cdot \alpha_k^{4/7}$$

Thus, combining (C.1) with Lemma C.2, we obtain

$$\epsilon_{k+1}^\theta \leq \prod_{j=0}^{k}(1 - \mu\alpha_j) \cdot \epsilon_0^\theta + (3M^2 + 6N\widetilde{H}^2) \cdot \sum_{l=0}^{k}\left[\alpha_l^2 \cdot \prod_{j=l+1}^{k}(1 - \mu\alpha_j)\right] \tag{D.3}$$

$$+ \widetilde{H}^2 \cdot \sum_{l=0}^{k}\left[(2\alpha_l^2 + \alpha_l/\mu) \cdot \epsilon_{l+1}^x \cdot \prod_{j=l+1}^{k}(1 - \mu\alpha_j)\right]$$

$$\leq 2\epsilon_0^\theta/\mu \cdot \alpha_k^{4/7} + (3M^2 + 6N\widetilde{H}^2 + 2\widetilde{H}^2) \cdot \sum_{l=0}^{k}\left[\alpha_l^2 \cdot \prod_{j=l+1}^{k}(1 - \mu\alpha_j)\right]$$

$$+ \widetilde{c} \cdot \widetilde{H}^2/\mu \cdot \sum_{l=0}^{k}\left[\alpha_l \beta_{l+1} \cdot \prod_{j=l+1}^{k}(1 - \mu\alpha_j)\right]$$

$$\leq 2\epsilon_0^\theta/\mu \cdot \alpha_k^{4/7} + 2 \cdot (3M^2 + 6N\widetilde{H}^2 + 2\widetilde{H}^2)/\mu \cdot \alpha_k^{4/7} + \widetilde{c} \cdot 2\beta\widetilde{H}^2/\alpha^{4/7}\mu^2 \cdot \alpha_k^{4/7}$$

$$= O(k^{-2/7}).$$

Here the third inequality follows from $\beta_k = \beta/\alpha^{4/7} \cdot \alpha_k^{4/7}$. Therefore, (D.2) and (D.3) conclude the proof of Theorem 5.5. $\qquad\square$

## D.1  Proof of Lemma D.1

*Proof.* We first show that

$$D_{\psi^i}\big(\widetilde{x}_*^i(\theta_k), x_{k+1}^i\big) = D_{\psi^i}\big(\widetilde{x}_*^i(\theta_k), \widetilde{x}_k^i\big) - 1/\beta_k^i \cdot \langle \widehat{v}_k^i, \widetilde{x}_*^i(\theta_k) - x_{k+1}^i \rangle - D_{\psi^i}(x_{k+1}^i, \widetilde{x}_k^i). \tag{D.4}$$

Since

$$x_{k+1}^i = \underset{x^i \in \mathcal{X}^i}{\mathrm{argmax}}\big\{\langle \widehat{v}_k^i, x^i \rangle - 1/\beta_k^i \cdot D_{\psi^i}(x_k^i, x^i)\big\},$$

we have the exact form of $x_{k+1}^i$ as

$$x_{k+1}^i \propto \widetilde{x}_k^i \cdot \exp\{\beta_k^i \cdot \widehat{v}_k^i\}. \tag{D.5}$$

By the definition of KL divergence, we have

$$D_{\psi^i}\big(\widetilde{x}_*^i(\theta_k), x_{k+1}^i\big) - D_{\psi^i}\big(\widetilde{x}_*^i(\theta_k), \widetilde{x}_k^i\big) \tag{D.6}$$

$$= \big\langle \log(\widetilde{x}_k^i/x_{k+1}^i), \widetilde{x}_*^i(\theta_k) - x_{k+1}^i \big\rangle - D_{\psi^i}(x_{k+1}^i, \widetilde{x}_k^i)$$

$$= \big\langle \log(\widetilde{x}_k^i/x_{k+1}^i) + \beta_k^i \cdot \widehat{v}_k^i, \widetilde{x}_*^i(\theta_k) - x_{k+1}^i \big\rangle - \beta_k^i \cdot \big\langle \widehat{v}_k^i, \widetilde{x}_*^i(\theta_k) - x_{k+1}^i \big\rangle - D_{\psi^i}(x_{k+1}^i, \widetilde{x}_k^i).$$

Let $Z_k^i = \sum_{j\in[d^i]}[\widetilde{x}_k^i]_j \cdot \exp\{1/\beta_k^i \cdot [\widehat{v}_k^i]_j\}$ be the normalization factor of the exact form of $x_{k+1}^i$ in (D.5). We have

$$\langle \log(\widetilde{x}_k^i/x_{k+1}^i) + \beta_k^i \cdot \widehat{v}_k^i, \widetilde{x}_*^i(\theta_k) - x_{k+1}^i\rangle \tag{D.7}$$
$$= \langle \log\widetilde{x}_k^i - \log x_{k+1}^i + \beta_k^i \cdot \widehat{v}_k^i, \widetilde{x}_*^i(\theta_k) - x_{k+1}^i\rangle$$
$$= \langle \log\widetilde{x}_k^i - \log\widetilde{x}_k^i - \beta_k^i \cdot \widehat{v}_k^i + \log Z_k^i + \beta_k^i \cdot \widehat{v}_k^i, \widetilde{x}_*^i(\theta_k) - x_{k+1}^i\rangle$$
$$= \log Z_{k+1}^i \cdot \langle \mathbf{1}_{d^i}, \widetilde{x}_*^i(\theta_k) - x_{k+1}^i\rangle = 0,$$

where the last equality follows from the fact that $\sum_{j\in[d^i]}[\widetilde{x}_*^i(\theta_k)]_j = \sum_{j\in[d^i]}[x_{k+1}^i]_j = 1$. Taking (D.7) into (D.6), we obtain

$$D_{\psi^i}\big(\widetilde{x}_*^i(\theta_k), x_{k+1}^i\big) = D_{\psi^i}\big(\widetilde{x}_*^i(\theta_k), \widetilde{x}_k^i\big) - \beta_k^i \cdot \langle \widehat{v}_k^i, \widetilde{x}_*^i(\theta_k) - x_{k+1}^i\rangle - D_{\psi^i}(x_{k+1}^i, \widetilde{x}_k^i),$$

which concludes the proof of (D.4).

Continuing from (D.4), we have

$$D_{\psi^i}\big(\widetilde{x}_*^i(\theta_k), x_{k+1}^i\big) = D_{\psi^i}\big(\widetilde{x}_*^i(\theta_k), \widetilde{x}_k^i\big) - \beta_k^i \cdot \langle \widehat{v}_k^i, \widetilde{x}_*^i(\theta_k) - x_{k+1}^i\rangle - D_{\psi^i}(x_{k+1}^i, \widetilde{x}_k^i)$$
$$\leq D_{\psi^i}\big(\widetilde{x}_*^i(\theta_k), \widetilde{x}_k^i\big) - \beta_k^i \cdot \langle \widehat{v}_k^i, \widetilde{x}_*^i(\theta_k) - \widetilde{x}_k^i\rangle$$
$$- \beta_k^i \cdot \langle \widehat{v}_k^i, \widetilde{x}_k^i - x_{k+1}^i\rangle - 1/2 \cdot \|x_{k+1}^i - \widetilde{x}_k^i\|_1^2,$$

where the second inequality follows from the fact that $D_{\psi^i}(x^i, x^{i\prime}) \geq 1/2 \cdot \|x^i - x^{i\prime}\|_1^2$. Taking the conditional expectation given $\mathcal{F}_k^x$, we obtain

$$\mathbb{E}\Big[D_{\psi^i}\big(\widetilde{x}_*^i(\theta_k), x_{k+1}^i\big) \,\Big|\, \mathcal{F}_k^x\Big] \leq D_{\psi^i}\big(\widetilde{x}_*^i(\theta_k), \widetilde{x}_k^i\big) - \beta_k^i \cdot \mathbb{E}\Big[\langle \widehat{v}_k^i, \widetilde{x}_*^i(\theta_k) - \widetilde{x}_k^i\rangle \,\Big|\, \mathcal{F}_k^x\Big] \tag{D.8}$$
$$+ \mathbb{E}\Big[-\beta_k^i \cdot \langle \widehat{v}_k^i, \widetilde{x}_k^i - x_{k+1}^i\rangle - 1/2 \cdot \|\widetilde{x}_k^i - x_{k+1}^i\|_1^2 \,\Big|\, \mathcal{F}_k^x\Big]$$
$$\leq D_{\psi^i}\big(\widetilde{x}_*^i(\theta_k), \widetilde{x}_k^i\big) - \langle \beta_k^i \cdot v_{\theta_k}^i(\widetilde{x}_k), \widetilde{x}_*^i(\theta_k) - x_*^i(\theta_k)\rangle$$
$$- \beta_k^i \cdot \langle v_{\theta_k}^i(\widetilde{x}_k), x_*^i(\theta_k) - \widetilde{x}_k^i\rangle + (\beta_k^i)^2/2 \cdot \mathbb{E}\big[\|\widehat{v}_k^i\|_\infty^2 \,\big|\, \mathcal{F}_k^x\big]$$
$$\leq D_{\psi^i}\big(\widetilde{x}_*^i(\theta_k), \widetilde{x}_k^i\big) - \beta_k^i \cdot \langle v_{\theta_k}^i(\widetilde{x}_k), x_*^i(\theta_k) - \widetilde{x}_k^i\rangle$$
$$+ (\beta_k^i)^2/2 \cdot \mathbb{E}\Big[\|\widehat{v}_k^i\|_\infty^2 + \big\|v_{\theta_k}^i(\widetilde{x}_k)\big\|_\infty^2 \,\Big|\, \mathcal{F}_k^x\Big] + 2\nu_k^2.$$

where the last inequality follows from Cauchy-Schwartz inequality and the fact that $\|\widetilde{x}_*^i(\theta_k) - x_*^i(\theta_k)\|_1 = \nu_k \cdot \|x_*^i(\theta_k) - \mathbf{1}_{d^i}/d^i\|_1 \leq 2\nu_k$. By the $\lambda$-strong variational stability of $x_*(\theta_k)$, we have

$$-\sum_{i\in\mathcal{N}} \lambda^i \cdot \langle v_{\theta_k}^i(\widetilde{x}_k), x_*^i(\theta_k) - \widetilde{x}_k^i\rangle \leq -\overline{D}_\psi\big(x_*(\theta_k), \widetilde{x}_k\big). \tag{D.9}$$

Moreover, by Assumption 5.3, we have

$$\mathbb{E}\Big[\|\widehat{v}_k^i\|_\infty^2 + \big\|v_{\theta_k}^i(\widetilde{x}_k)\big\|_\infty^2 \,\Big|\, \mathcal{F}_k^x\Big]$$
$$\leq \mathbb{E}\Big[2\big\|\widehat{v}_k^i - v_{\theta_k}^i(\widetilde{x}_k)\big\|_\infty^2 + 3\big\|v_{\theta_k}^i(\widetilde{x}_k)\big\|_\infty^2 \,\Big|\, \mathcal{F}_k^x\Big] \leq 2\delta_u^2 + 3\big\|v_{\theta_k}^i(\widetilde{x}_k)\big\|_\infty^2,$$

summing up which gives

$$\sum_{i\in\mathcal{N}}(\lambda^i)^2 \cdot \mathbb{E}\Big[\|\widehat{v}_k^i\|_\infty^2 + \big\|v_{\theta_k}^i(\widetilde{x}_k)\big\|_\infty^2 \,\Big|\, \mathcal{F}_k^x\Big] \tag{D.10}$$
$$\leq 2\delta_u^2 \cdot \sum_{i\in\mathcal{N}}(\lambda^i)^2 + 6 \cdot \sum_{i\in\mathcal{N}}(\lambda^i)^2 \cdot \left(\big\|v_{\theta_k}^i(\widetilde{x}_k) - v_{\theta_k}^i(x_*(\theta_k))\big\|_\infty^2 + \big\|v_{\theta_k}^i(x_*(\theta_k))\big\|_\infty^2\right)$$
$$\leq (2\delta_u^2 + 6V_*^2) \cdot \sum_{i\in\mathcal{N}}(\lambda^i)^2 + 6NH_u^2 \cdot \sum_{i\in\mathcal{N}}(\lambda^i)^2 \cdot \overline{D}_\psi\big(x_*(\theta_k), \widetilde{x}_k\big)$$
$$= (2\delta_u^2 + 6V_*^2) \cdot \|\lambda\|_2^2 + 6NH_u^2\|\lambda\|_2^2 \cdot \overline{D}_\psi\big(x_*(\theta_k), \widetilde{x}_k\big),$$

where the second inequality follows from $\|v_\theta(x_*(\theta))\|_\infty \leq V_*$, and the third inequality follows from Assumption 5.1. Thus, taking (D.9) and (D.10) into (D.8), we obtain for $6NH_u^2\|\lambda\|_2^2\beta_k^2 \leq \beta_k$ (i.e.,

$\beta \leq 1/6NH_u^2\|\lambda\|_2^2$) that,

$$\mathbb{E}\Big[\overline{D}_\psi\big(\widetilde{x}_*(\theta_k), x_{k+1}\big) \,\Big|\, \mathcal{F}_k^x\Big]$$

$$\leq \overline{D}_\psi\big(\widetilde{x}_*(\theta_k), \widetilde{x}_k\big) - \big(\beta_k - 3NH_u^2\|\lambda\|_2^2\beta_k^2\big) \cdot \overline{D}_\psi\big(x_*(\theta_k), \widetilde{x}_k\big) + (\delta_u^2 + 3V_*^2) \cdot \|\lambda\|_2^2\beta_k^2 + 2N\nu_k^2$$

$$\leq (1 - \beta_k/2) \cdot \overline{D}_\psi\big(\widetilde{x}_*(\theta_k), \widetilde{x}_k\big) + (\delta_u^2 + 3V_*^2) \cdot \|\lambda\|_2^2\beta_k^2 + N\beta_k\nu_k \log(1/\nu_k) + 2N\nu_k^2,$$

where the second inequality follows from Lemma E.3. By Lemma E.3, we further have for $\nu_k \leq O(1/k)$,

$$\mathbb{E}\Big[\overline{D}_\psi\big(\widetilde{x}_*(\theta_k), \widetilde{x}_{k+1}\big) \,\Big|\, \mathcal{F}_k^x\Big] - 2N\nu_{k+1} \tag{D.11}$$

$$\leq \mathbb{E}\Big[\overline{D}_\psi\big(\widetilde{x}_*(\theta_k), x_{k+1}\big) \,\Big|\, \mathcal{F}_k^x\Big]$$

$$\leq (1 - \beta_k/2) \cdot \overline{D}_\psi\big(\widetilde{x}_*(\theta_k), \widetilde{x}_k\big) + (\delta_u^2 + 3V_*^2) \cdot \|\lambda\|_2^2\beta_k^2 + N\beta_k\nu_k \log(1/\nu_k) + 2N\nu_k^2,$$

By Lemma E.2, we have for any $\gamma > \max\{1, 1/\nu_k^2\}$,

$$\overline{D}_\psi\big(\widetilde{x}_*(\theta_k), \widetilde{x}_k\big) - \Big(1 + \frac{1}{\gamma}\Big) \cdot \overline{D}_\psi\big(\widetilde{x}_*(\theta_{k-1}), \widetilde{x}_k\big) \tag{D.12}$$

$$\leq \frac{(1+\gamma)^2/\nu_k^2 - (1+\gamma)}{2\gamma} \cdot \big\|\widetilde{x}_*(\theta_k) - \widetilde{x}_*(\theta_{k-1})\big\|_1^2$$

$$\leq \frac{(1+\gamma)^2/\nu_k^2 - (1+\gamma)}{2\gamma} \cdot \|\theta_k - \theta_{k-1}\|_2^2 \leq \frac{(1+\gamma)^2/\nu_k^2 - (1+\gamma)}{2\gamma} \cdot \widetilde{H}_*^2\alpha_{k-1}^2 \cdot \|\widetilde{\nabla}f_{k-1}\|_2^2,$$

where the second inequality follows from Lemma B.3. Taking (D.12) into (D.11) and choosing $\gamma = (4 - 2\beta_k)/\beta_k$, we obtain

$$\mathbb{E}\Big[\overline{D}_\psi\big(\widetilde{x}_*(\theta_k), \widetilde{x}_{k+1}\big) \,\Big|\, \mathcal{F}_k^x\Big] \tag{D.13}$$

$$\leq (1 - \beta_k/2) \cdot \Big(1 + \frac{1}{\gamma}\Big) \cdot \overline{D}_\psi\big(x_*(\theta_{k-1}), x_k\big) + (\delta_u^2 + 3V_*^2) \cdot \|\lambda\|_2^2\beta_k^2 + N\beta_k\nu_k \log(1/\nu_k)$$

$$+ 2N \cdot (\nu_k^2 + \nu_{k+1}) + (1 - \beta_k/2) \cdot \frac{(1+\gamma)^2/\nu_k^2 - (1+\gamma)}{2\gamma} \cdot \widetilde{H}_*^2\alpha_{k-1}^2 \cdot \mathbb{E}\big[\|\widetilde{\nabla}f_{k-1}\|_2^2 \,\big|\, \mathcal{F}_k^x\big]$$

$$= (1 - \beta_k/4) \cdot \overline{D}_\psi\big(x_*(\theta_{k-1}), x_k\big) + (\delta_u^2 + 3V_*^2) \cdot \|\lambda\|_2^2\beta_k^2 + N\beta_k\nu_k \log(1/\nu_k)$$

$$+ 2N \cdot (\nu_k^2 + \nu_{k+1}) + (1 - \beta_k/4) \cdot \frac{(1+\gamma)/\nu_k^2 - 1}{2} \cdot \widetilde{H}_*^2\alpha_{k-1}^2 \cdot \mathbb{E}\big[\|\widetilde{\nabla}f_{k-1}\|_2^2 \,\big|\, \mathcal{F}_k^x\big].$$

By Assumption 5.3, we have

$$\mathbb{E}\big[\|\widetilde{\nabla}f_{k-1}\|_2^2 \,\big|\, \mathcal{F}_{k-1}^\theta\big] \tag{D.14}$$

$$\leq 3 \cdot \mathbb{E}\bigg[\Big\|\widetilde{\nabla}f\big(\theta_{k-1}, \widetilde{x}_k\big) - \widetilde{\nabla}f\big(\theta_{k-1}, \widetilde{x}_*(\theta_{k-1})\big)\Big\|_2^2 \,\bigg|\, \mathcal{F}_{k-1}^\theta\bigg]$$

$$+ 3 \cdot \mathbb{E}\bigg[\Big\|\widetilde{\nabla}f\big(\theta_{k-1}, \widetilde{x}_*(\theta_{k-1})\big) - \nabla f_*(\theta_{k-1})\Big\|_2^2 \,\bigg|\, \mathcal{F}_{k-1}^\theta\bigg] + 3 \cdot \mathbb{E}\big[\|\nabla f_*(\theta_{k-1})\|_2^2 \,\big|\, \mathcal{F}_{k-1}^\theta\big]$$

$$\leq 3\widetilde{H}^2 \cdot \overline{D}_\psi\big(\widetilde{x}_*(\theta_{k-1}), \widetilde{x}_k\big) + 3\widetilde{H}^2 \cdot \overline{D}_\psi\big(x_*(\theta_{k-1}), \widetilde{x}_*(\theta_{k-1})\big) + 3M^2$$

$$\leq 3\widetilde{H}^2 \cdot \overline{D}_\psi\big(\widetilde{x}_*(\theta_{k-1}), \widetilde{x}_k\big) + 3M^2 + 6N\widetilde{H}^2\nu_{k-1},$$

where the third inequality follows from Assumptions 5.2 and 5.3, and the last inequality follows from Lemma E.3. Taking (D.14) into (D.13) and taking expectation on both sides, we obtain

$$\widetilde{\epsilon}_{k+1}^x \leq \Big\{1 - \beta_k/4 + 3\big[(1+\gamma)/\nu_k^2 - 1\big]/2 \cdot \widetilde{H}^2\widetilde{H}_*^2\alpha_{k-1}^2\Big\} \cdot \widetilde{\epsilon}_k^x + (\delta_u^2 + 3V_*^2) \cdot \|\lambda\|_2^2\beta_k^2$$

$$+ (1 - \beta_k/4) \cdot \frac{(1+\gamma)/\nu_k^2 - 1}{2} \cdot \widetilde{H}_*^2\alpha_{k-1}^2 \cdot \big(2M^2 + 6N\widetilde{H}^2\nu_{k-1}\big)$$

$$- N\beta_k\nu_k \log\nu_k + 2N \cdot (\nu_{k+1} + \nu_k^2). \tag{D.15}$$

With $\gamma = (4\beta_k - 2)/\beta_k$, $\alpha_k = \alpha/(k+1)$, $\beta_k = \beta/(k+1)^{2/7}$, $\nu_k = 1/(k+1)^{4/7}$, and $\alpha/\beta^{3/2} \leq 1/7\widetilde{H}\widetilde{H}_*$, we have

$$3\big[(1+\gamma)/\nu_k^2 - 1\big]/2 \cdot \widetilde{H}^2 \widetilde{H}_*^2 \alpha_{k-1}^2 = 3\big[(4/\beta_k - 1)/\nu_k^2 - 1\big]/2 \cdot \widetilde{H}^2 \widetilde{H}_*^2 \alpha_{k-1}^2$$
$$\leq 6\widetilde{H}^2 \widetilde{H}_*^2 \alpha_k^2/\beta_k \nu_k^2 \leq \beta_k^2/8 \leq \beta_k/8,$$

taking which into (D.15), we obtain

$$\widetilde{\epsilon}_{k+1}^x \leq (1 - \beta_k/8) \cdot \widetilde{\epsilon}_k^x + (\delta_u^2 + 3V_*^2) \cdot \|\lambda\|_2^2 \beta_k^2 - N\beta_k\nu_k \log\nu_k + 2N \cdot (\nu_{k+1} + \nu_k^2)$$
$$+ \beta_k^2 \cdot (M^2 + 3N\widetilde{H}^2\nu_{k-1})/4\widetilde{H}^2$$
$$\leq (1 - \beta_k/8) \cdot \epsilon_k^x + (\delta_u^2 + 3V_*^2) \cdot \|\lambda\|_2^2 \beta_k^2 - N\beta_k\nu_k \log\nu_k + 2N \cdot (\nu_{k+1} + \nu_k^2)$$
$$+ \beta_k^2 \cdot (M^2 + 3N\widetilde{H}^2)/4\widetilde{H}^2. \tag{D.16}$$

Recursively applying (D.16), we obtain

$$\epsilon_{k+1}^x \leq \epsilon_0^x \cdot \prod_{j=0}^{k}(1 - \beta_j/8) + \big[(\delta_u^2 + 3V_*^2) \cdot \|\lambda\|_2^2 + (M^2 + 3N\widetilde{H}^2)/4\widetilde{H}^2\big] \cdot \left[\sum_{l=0}^{k}\beta_l^2 \cdot \prod_{j=l+1}^{k}(1 - \beta_j/8)\right]$$
$$+ N \cdot \left[\sum_{l=0}^{k}(-\beta_l\nu_l \log\nu_l + 2\nu_{l+1} + 2\nu_l^2) \cdot \prod_{j=l+1}^{k}(1 - \beta_j/8)\right].$$

$\square$

## D.2   Proof of Lemma D.2

*Proof.* Applying (D.14) to (C.12) and taking expectation on both sides, we get

$$\epsilon_{k+1}^\theta \leq (1 - \mu\alpha_k) \cdot \epsilon_k^\theta + (3M^2 + 6N\widetilde{H}^2\nu_k) \cdot \alpha_k^2 + \widetilde{H}^2 \cdot (3\alpha_k^2 + \alpha_k/\mu) \cdot \epsilon_{k+1}^x$$
$$\leq (1 - \mu\alpha_k) \cdot \epsilon_k^\theta + (3M^2 + 6N\widetilde{H}^2) \cdot \alpha_k^2 + \widetilde{H}^2 \cdot (3\alpha_k^2 + \alpha_k/\mu) \cdot \epsilon_{k+1}^x, \tag{D.17}$$

where the second inequality follows from $\nu_k \leq 1$. Recursively applying (D.17), we obtain

$$\epsilon_{k+1}^\theta \leq \prod_{j=0}^{k}(1 - \mu\alpha_j) \cdot \epsilon_0^\theta + (3M^2 + 6N\widetilde{H}^2) \cdot \sum_{l=0}^{k}\left[\alpha_l^2 \cdot \prod_{j=l+1}^{k}(1 - \mu\alpha_j)\right]$$
$$+ \widetilde{H}^2 \cdot \sum_{l=0}^{k}\left[(2\alpha_l^2 + \alpha_l/\mu) \cdot \epsilon_{l+1}^x \cdot \prod_{j=l+1}^{k}(1 - \mu\alpha_j)\right].$$

$\square$

# E   Properties of the Bregman Divergence

The following lemma is used in the analysis of unconstrained games.

**Lemma E.1.** *Let $\psi(\cdot)$ be $1$-strongly convex with respect to the norm $\|\cdot\|$. Assume that (4.2) holds. We have for any $\gamma > H_\psi^2 \geq 1$,*

$$D_\psi(x, z) - \left(1 + \frac{1}{\gamma}\right) \cdot D_\psi(y, z) \leq \frac{H_\psi^2 \cdot (1+\gamma)^2 - (1+\gamma)}{2\gamma} \cdot \|x - y\|^2.$$

*Proof.* By the definition of Bregman divergence, we have for any $\gamma > 0$,

$$D_\psi(x, z) - D_\psi(y, z) = \psi(x) - \langle\nabla\psi(z), x - z\rangle - \psi(y) + \langle\nabla\psi(z), y - z\rangle$$
$$= -D_\psi(x, y) + \langle\nabla\psi(z) - \nabla\psi(x), y - x\rangle$$
$$\leq -1/2 \cdot \|x - y\|^2 + \big\|\nabla\psi(z) - \nabla\psi(x)\big\|_* \cdot \|y - x\|,$$

where the inequality follows from 1-strong convexity of $\psi(\cdot)$ and the Cauchy-Schwartz inequality. By (4.2), we further have

$$D_\psi(x,z) - D_\psi(y,z) \leq -1/2 \cdot \|x-y\|^2 + H_\psi \cdot \|z-x\| \cdot \|y-x\|,$$

$$\leq \frac{1}{2(1+\gamma)} \cdot \|x-z\|^2 + \frac{H_\psi^2 \cdot (1+\gamma) - 1}{2} \cdot \|x-y\|^2$$

$$\leq \frac{1}{1+\gamma} \cdot D_\psi(x,z) + \frac{H_\psi^2 \cdot (1+\gamma) - 1}{2} \cdot \|x-y\|^2, \tag{E.1}$$

where the second inequality follows from 1-strong convexity of $\psi(\cdot)$. Rearranging the terms in (E.1), we finish the proof of Lemma E.1. $\qquad\square$

The following two lemmas are involved in the analysis of simplex constrained games.

**Lemma E.2.** *Let $\psi(\cdot)$ be the Shannon entropy. We have for any $\gamma_k > \max\{1, 1/\nu_k^2\}$,*

$$D_{\psi^i}\big(\widetilde{x}_*^i(\theta_k), \widetilde{x}_k^i\big) - \left(1 + \frac{1}{\gamma}\right) \cdot D_{\psi^i}\big(\widetilde{x}_*^i(\theta_{k-1}), \widetilde{x}_k^i\big) \leq \frac{(1+\gamma)^2/\nu_k^2 - (1+\gamma)}{2\gamma} \cdot \big\|\widetilde{x}_*^i(\theta_k) - \widetilde{x}_*^i(\theta_{k-1})\big\|_1^2,$$

*for $\{\widetilde{x}_k\}_{k\geq 0}$ generated by Algorithm 2.*

*Proof.* For the Shannon entropy $\psi(\cdot)$, we have $\nabla_{x_j}\psi(\widetilde{x}) = 1 + \log[\widetilde{x}]_j$, which gives

$$\left|\nabla_{[x^i]_j}\psi\big(\widetilde{x}_*^i(\theta_k)\big) - \nabla_{[x^i]_j}\psi\big(\widetilde{x}_k^i\big)\right| = \left|\log\big[\widetilde{x}_*^i(\theta_k)\big]_j - \log[\widetilde{x}_k^i]_j\right| \leq 1/\nu_k \cdot \left|\big[\widetilde{x}_*^i(\theta_k)\big]_j - [\widetilde{x}_k^i]_j\right|.$$

Thus, we have

$$\left\|\nabla\psi\big(\widetilde{x}_*^i(\theta_k)\big) - \nabla\psi\big(\widetilde{x}_k^i\big)\right\|_\infty \leq 1/\nu_k \cdot \big\|\widetilde{x}_*^i(\theta_k) - \widetilde{x}_k^i\big\|_1.$$

Replacing $H_\psi$ with $1/\nu_k$ in the proof of Lemma E.1, we conclude the proof of Lemma E.2. $\qquad\square$

**Lemma E.3.** *Let $\{x_k\}_{k\geq 0}$ and $\{\widetilde{x}_k\}_{k\geq 0}$ be the sequences of strategy profiles generated by Algorithm 2 with $\nu_k \leq O(1/k)$. We have*

$$\overline{D}_\psi\big(\widetilde{x}_*(\theta_k), \widetilde{x}_k\big) - \overline{D}_\psi\big(\widetilde{x}_*(\theta_k), x_k\big) \leq 2N\nu_k, \tag{E.2}$$

$$\overline{D}_\psi\big(\widetilde{x}_*(\theta_k), \widetilde{x}_{k+1}\big) - \overline{D}_\psi\big(\widetilde{x}_*(\theta_k), x_{k+1}\big) \leq 2N\nu_{k+1}, \tag{E.3}$$

$$\overline{D}_\psi\big(\widetilde{x}_*(\theta_k), \widetilde{x}_k\big) - \overline{D}_\psi\big(x_*(\theta_k), \widetilde{x}_k\big) \leq 2N\nu_k \log(1/\nu_k). \tag{E.4}$$

*Proof.* By the definition of KL divergence, we have

$$D_\psi\big(\widetilde{x}_*^i(\theta_k), \widetilde{x}_{k+1}^i\big) - D_\psi\big(\widetilde{x}_*^i(\theta_k), x_{k+1}^i\big) = \big\langle\widetilde{x}_*^i(\theta_k), \log(x_{k+1}^i/\widetilde{x}_{k+1}^i)\big\rangle. \tag{E.5}$$

By (4.6), we have for all $\nu_k \leq O(k^{-1})$ that

$$\log\frac{[x_{k+1}^i]_j}{[\widetilde{x}_{k+1}^i]_j} = \log\left\{\frac{[x_{k+1}^i]_j}{(1-\nu_{k+1})\cdot[x_{k+1}^i]_j + \nu_{k+1}/d^i}\right\}$$

$$= \log\left\{1 + \frac{\nu_{k+1}\cdot\big([x_{k+1}^i]_j - 1/d^i\big)}{(1-\nu_{k+1})\cdot[x_{k+1}^i]_j + \nu_{k+1}/d^i}\right\} \leq 2\nu_{k+1}. \tag{E.6}$$

Taking (E.6) into (E.5), we obtain

$$\overline{D}_\psi\big(\widetilde{x}_*(\theta_k), \widetilde{x}_{k+1}\big) - \overline{D}_\psi\big(\widetilde{x}_*(\theta_k), x_{k+1}\big) \leq \sum_{i\in\mathcal{N}}\big\langle\widetilde{x}_*^i(\theta_k), \mathbf{1}_{d^i}\big\rangle \cdot 2\nu_{k+1} = 2N\nu_{k+1},$$

which concludes the proof of (E.2). Similar arguments also yields (E.3). Also, we have

$$D_{\psi^i}\big(\widetilde{x}_*^i(\theta_k), \widetilde{x}_k^i\big) - D_{\psi^i}\big(x_*^i(\theta_k), \widetilde{x}_k^i\big)$$

$$= \Big\langle\widetilde{x}_*^i(\theta_k), \log\big(\widetilde{x}_*^i(\theta_k)/\widetilde{x}_k^i\big)\Big\rangle - \Big\langle x_*^i(\theta_k), \log\big(x_*^i(\theta_k)/\widetilde{x}_k^i\big)\Big\rangle$$

$$= \Big\langle\widetilde{x}_*^i(\theta_k) - x_*^i(\theta_k), \log\big(\widetilde{x}_*^i(\theta_k)/\widetilde{x}_k^i\big)\Big\rangle - D_{\psi^i}\big(x_*^i(\theta_k), \widetilde{x}_*^i(\theta_k)\big)$$

$$\leq \big\|\widetilde{x}_*^i(\theta_k) - x_*^i(\theta_k)\big\|_1 \cdot \Big\|\log\big(\widetilde{x}_*^i(\theta_k)/\widetilde{x}_k^i\big)\Big\|_\infty \leq 2\nu_k\log(1/\nu_k),$$

summing up which for $i\in\mathcal{N}$ gives (E.4). $\qquad\square$

# F Extensions to Games with Multiple Equilibria

In this section, we provide more details on how to extend our algorithms to games with multiple equilibria. As discussed in Section 6, the only challenging case is when the function $v_{\theta(x)}$ is monotone but *not* strongly monotone, so that the equilibrium set is a convex and closed region.

**Example F.1.** We consider a specific routing game (see Example 3.2). Specifically, we study a simple network as shown in Figure F.1. Suppose that the number of nonatomic agents aiming to travel from node 1 to node 4 is $d = 10$. The costs for using the 4 edges are given by $t^1(x^1) = 4 + (x^1)^4$,

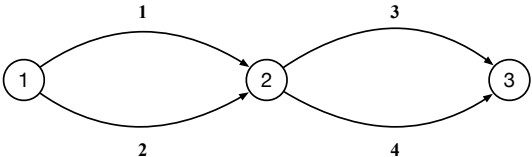

Figure F.1: A 3-node-4-link network.

$t^2(x^2) = 20 + 5(x^2)^4$, $t^3(x^3) = 1 + 30(x^3)^4$ and $t^4(x^4) = 30 + (x^4)^4$, respectively. There are 4 paths connecting node 1 and node 4 (path 1 uses link 1 and link 3, path 2 uses link 2 and link 4, path 3 uses link 1 and link 4, path 4 uses link 2 and 3). Let $q = (q^a)_{a=1}^4$ be the number of agents using each path and $c(q) = (c^a(q))_{a=1}^4$ be cost of using each path. Then we have

$$c(q) = \Delta^{\mathsf{T}} t(x),$$

where $\Delta = [1,0,1,0;0,1,0,1;1,0,0,1;0,1,1,0]$. Therefore, we have $\nabla c(q) = \Delta^{\mathsf{T}} \mathrm{Diag}(t'(x))\Delta$, which is semi-positive definite but not positive definite because the matrix $\Delta$ is singular. Under this setting, the set of equilibria to this routing game can be written as

$$\{f^* \geq 0 : \Delta f^* = x^*\}, \quad \text{where } x^* = [6,4,3,7]^{\mathsf{T}}. \tag{F.1}$$

Yet, we can simply add a regularizer $\eta^i \cdot (\log(x^i + \epsilon) + 1)$ to each $v_\theta^i(x)$. We note that the Jacobian of $v_\theta^i(x) + \eta^i \cdot (\log(x^i + \epsilon) + 1)$ is $\nabla v_\theta^i(x) + \eta^i \cdot \mathrm{Diag}(1/(x^i + \epsilon))$. The unique equilibrium then would be strongly stable (see Lemma B.2). Meanwhile, if $\epsilon > 0$, the Jacobian would also be upper-bounded, hence the function $v_\theta^i(x) + \eta^i \cdot (\log(x^i + \epsilon) + 1)$ is still Lipschitz continuous. We summarize a few properties of this regularizer. First, as long as $\eta^1 > 0$ and $\epsilon > 0$, all conditions in Assumption 5.1 would be satisfied. Seond, if $\eta^1 > 0$ and $\epsilon = 0$, the resulting equilibrium is often referred to as a quantal response (response) equilibrium, would is more realistic than a Nash (Wardrop) equilibrium if we believe that human beings are not always fully rational [42, 1, 33]. Third, if $\eta^1 > 0$ and $\epsilon > 0$, the resulting equilibrium would be a "smoothed" quantal response equilibrium. It is close to the quantal response equilibrium when $\epsilon$ is sufficiently small, but it preserves the Lipschitz continuity.