# OpenReview forum: "Inducing Equilibria via Incentives: Simultaneous Design-and-Play Ensures Global Convergence"
_NeurIPS.cc/2022/Conference — NeurIPS 2022 Accept_

### Official Review · Reviewer_ZpdR · 2022-07-09

**Rating:** 5
**Confidence:** 3
**Soundness:** 3 good
**Presentation:** 2 fair
**Contribution:** 3 good

**Summary:**

This paper claims to introduce algorithms for incentive design in convex games. In particular, the incentive designer is faced with selecting a parameter $\theta$ such that the resulting game has an equilibrium $x_*(\theta)$ that is desirable for the designer. The authors state some preliminary results about such games, and then give an algorithm that attempts to find an optimal parameter $\theta$, prove properties of their algorithms, and give some experimental evidence of their effectiveness in practice (appendix).

**Questions:**

A few minor points about the exposition (atomic and nonatomic games):

1. Why separate "atomic games" from "nonatomic games"? It seems to me that a nonatomic game is simply an atomic game over simplices with multilinear utility functions.
2. Also, why define one in terms of utility and one in terms of cost? To me this only introduces extra needless confusion.
3. Isn't "nonatomic game" just a normal-form game here? Why bother introducing "continuum of players", and why not use the usual language of normal-form games?

Miscellaneous things:

1. Throughout the paper, "quantum" should probably be "quantal", as in "quantal response equilibrium". I've never heard of that referred to as "quantum equilibrium".
1. Line 145: since the utility is being maximized, $u^i$ should be concave, not convex

**Limitations:**

The authors have adequately addressed limitations, modulo concerns I have already stated above.

**Strengths And Weaknesses:**

The problem and analysis presented in the paper are interesting. However, I have some concerns, mostly pertaining to setting this paper within the field of computational game theory.

Perhaps most notably, I do not think the authors have done a sufficient job relating this paper to prior work in computational game theory. For example, the techniques of this paper are based on online learning--Algorithms 1 & 2 are, modulo some careful tuning of step sizes, essentially "every player run online mirror descent", which is a very well-known idea in computational game theory. However, the authors do not discuss this connection at all. Further, the "bilevel optimization problem" is essentially a Stackelberg game with a single leader and multiple followers, and I believe there should be a concrete connection drawn to that literature. For example, if $n=1$, then it is known that equilibria are easy to find even in general normal-form games, by a standard linear programming algorithm.

I would also like to see a much more in-depth discussion of the Assumptions 5.1-5.3 as they pertain to useful classes of games. For example:
1. The monotonicity/strong stability condition (e.g., Assumption 5.1, Section 6 Case II) seems *very* strong. I understand that this is the only thing preventing the paper from breaking long-standing assumptions in complexity theory (e.g., implying a polynomial-time algorithm for Nash in normal-form games), but I feel that the authors should take more time to discuss intuition for what these assumptions mean in practice. For example: what, if anything, does this paper say about normal-form ("nonatomic", in this paper's language) games?
1. The convexity of $f_*(\theta)$ is perhaps too strong an assumption. The authors admit this, but that does not make it less bothersome.
1. Can the bounded variance in Assumption 5.3 be replaced with an $x^i$-dependent variance, as is usual in the online learning literature (see, e.g., the analysis of Exp3)?

---

> ### Author Response · Authors · 2022-08-02
> **Response to Reviewer ZpdR**
>
> Thanks for providing comments and suggestions. Below are our responses to your questions.
>
> **1. How our work is related to the online mirror descent (MD) algorithm.**
>
> To start with, our algorithm is definitely **not** a direct application of MD just "modulo some careful tuning of step sizes". The incentive design problem is a bilevel program, which is equivalent to a Stackelberg game.
>
>
> For example, let's just consider a simple Stackelberg game with one leader and one follower, whose cost functions are $f(x, y)$ and $g(x, y)$, respectively ($x$ is the leader's action while $y$ is the follower's decision). The Stackelberg game then can be framed as
>
> $$
> \min_x f(x, y), \quad \text{subject to} \ y \in \arg \min_{y} g(x, y).
> $$
>
> To apply MD and invoke classic results to understand its convergence, we have two natural thoughts:
>
> -   Use the gradient $\nabla_x \\ f(x_k, y_k)$ and $\nabla_y \\ f(x_k, y_k)$ to update the leader's action at each iteration $k$. In this case, the convergence of MD is well-understood (as a classic result in "online learning in games"). However, $\nabla_x \\ f(x_k, y_k)$ is not the true gradient of the leader's cost because the dependence of $y$ on $x$ is ignored.
>
> -   Use the true gradient to update the leader's action at each iteration $k$, which --- according to the chain rule --- reads
>
>     $$
>     \nabla_x \\  f(x_k, y^*(x_k)) + \frac{\partial y^*(x_k)}{\partial x_k} \cdot \nabla_y \\ f(x_k, y^*(x_k)),
>     $$
>
>     If so, the convergence of the algorithm would also become easy to follow (as a classic result in "online optimization"). However, it requires repeatedly calculating the best response Jacobian $y^*(x_k)$ at each iteration $k$, which is a computational burden.
>
>
> In a nutshell, the direct application of MD is either "false" or "inefficient". That's why we need to devise **new** algorithms for solving such problems. **Our algorithm only adopts the overall structure of MD; it does not use the gradient of any actual cost functions to update the leader's cost**. Hence,  the convergence analysis is extremely difficult.
>
>
> **2. Assumptions**
>
> - Our assumption imposed on the lower-level problem is common in computational game theory (e.g., [1, 2]).
>
> [1] Zhou, Zhengyuan, et al. "Mirror descent learning in continuous games." 2017 IEEE 56th Annual Conference on Decision and Control (CDC). IEEE, 2017.
>
> [2] Mertikopoulos, Panayotis, and Zhengyuan Zhou. "Learning in games with continuous action sets and unknown payoff functions." Mathematical Programming 173.1 (2019)
>
> - As it is widely recognized that nonlinear MPECs can only be solved locally, we directly assume that the objective function is strongly convex.
>
> - The assumption on the variance bounds can be generalized to be player-specific, which does not affect the algorithm design, and only requires minimal adaptation in the analysis.
>
>
>
> **3. Relation with Stackelberg game**
>
> The incentive design problem, of course, can be interpreted as a Stackelberg game. But a Stackelberg game with one follower is essentially a bilevel program, while a Stackelberg game with multiple followers is essentially an MPEC. Hence, in Section 2, we respectively discuss bilevel optimization and MPEC for clarity.
>
> We also want clarify that in a Stackelberg game, if the followers' problem can be formulated as a linear program, the overall problem would become NP-hard [3]. Hence, it is not the focus of the present study; the variational stability condition imposed on the lower level could prevent this case.
>
> [3] Bard, Jonathan F. "Some properties of the bilevel programming problem." Journal of optimization theory and applications 68.2 (1991):
>
> **4. Atomic games v.s. nonatomic games**
>
> This classification is a common practice. For example, in the book *Algorithmic Game Theory*, the difference between atomic games and nonatomic games is carefully discussed.
>
> [4] Nisan, N., Roughgarden, T., Tardos, E., & Vazirani, V. V. (Eds.). (2007). Algorithmic Game Theory. Cambridge University Press.
>
> The solution concept for the atomic game is named after Nash while the solution concept for the nonatomic game is named after Wardrop.  A nonatomic game (e.g., routing game, congestion game, population game) is not a special normal-form game.

---

> > ### Comment · Reviewer_ZpdR · 2022-08-06
> > **Response to response**
> >
> > The response has addressed some of my concerns, and as such I raise my score.
> >
> > On nonatomic games: Gah, I had missed the point in the formulation that the cost functions $c^{ia}(q)$ don't really need to have any structure; in particular, they can depend on $q^i$ and be nonlinear. I am not used to this formulation. That is a somewhat embarrassing mistake on my part; I am sorry.
> >
> > Your response contains two methods of attempting mirror descent in games. I am referring to the former, which, as you point out, is well analyzed in the online learning literature, and to which I find this paper's algorithms quite similar. In particular, the key differences seems to be (1) the two-timescale stepsizing (which is clearly critical, as the problem is bilevel), and (2) the definition of $\tilde\nabla$. It is unclear to me how important the latter is: for example, does the algorithm fail to converge if one simply uses $\nabla_\theta f(\theta_k, x_{k+1})$ in place of $\tilde \nabla_\theta f(\theta_k, x_{k+1})$, all else kept equal? I would appreciate some discussion regarding this point, and possibly an explicit counterexample if the convergence indeed fails.
> >
> > The reason I do not raise further is that I believe that the present paper still lacks critical intuition and connection to past work. In particular, I would strongly suggest that the authors take some time to discuss how this paper relates (or does not relate) to the literature on normal-form games, in particular on online mirror descent and Stackelberg equilibria, as per the current conversation. As that literature is closer to home for many in this field (myself included), it would serve as a reasonable building block for intuition regarding the contributions of the paper.
> >
> > Also, a minor note I forgot to mention in the original review: it seems needlessly cumbersome to define atomic games in terms of *utilities* and nonatomic games in terms of *costs*; it just makes you have to write "utilities/costs" later in the paper, for example. While it obviously makes no mathematical difference, I'd prefer sticking to one or the other.

---

> > > ### Author Response · Authors · 2022-08-07
> > > **Further Response**
> > >
> > > We sincerely thank the reviewer for raising the rating of our work. For your remaining questions, we still hope to make some explanation.
> > >
> > > ##### **1. Use $\nabla_{\theta }\\ f(\theta_k, x_{k + 1})$ to update the incentives**
> > >
> > > We hope to clarify that directly using $\nabla_{\theta }\\ f(\theta_k, x_{k + 1})$ to update the incentives would not affect the convergence of the algorithm. **The caveat is that the algorithm would converge to a wrong solution**.
> > >
> > > **Example 1**. Let's consider the classic Stackelberg duopoly model. Regardless of some details, it can be formulated as the following bilevel optimization problem.
> > > \begin{equation}
> > > \min_{1 \geq \theta \geq 0} l(\theta, x), \quad \text{s.t.} \\ y^* = \arg \min_{1 \geq x \geq 0} f(\theta, x),
> > > \end{equation}
> > > where $l(theta, x) = - \theta (1 - \theta - x)$ and $f(\theta, x) = -x (1 - \theta - y)$.
> > >
> > > It can be easily checked that for any $\theta \in [0, 1]$, the lower-level problem admits an analytic solution $x^* = (1 - \theta) / 2$. Replacing $x^* = (1 - \theta) / 2$ in the upper level, we can eventually obtain that $\theta^* = 1/2$ and $x^* = 1/4$, at which we have $l(\theta^*, x^*) = -1/8$.
> > >
> > > But if we directly apply the projected gradient descent to this problem (the first approach mentioned in our previous response), i.e.,
> > > - The follower iteratively update $x^k$ based on $\nabla_y \\ f(\theta^k, x^k)$,
> > > - The leader iteratively updates $\theta^k$ based on $\nabla_x \\ l(\theta^k, x^{k + 1})$,
> > >
> > > Then it can be easily verified that the algorithm would converge to $\bar \theta = 1/3$ and $\bar x = 1/3$, at which we have $l(\bar \theta, \bar x) = - 1/9 > - 1/8$. Hence, it is not the correct solution to the original problem.
> > >
> > > **Example 2.** Let's move to the second-best congestion-pricing problem (the numerical example that we provided in Appendix F.2). In this example, the incentive designer's objective function is described as the total travel delay experienced by the travelers. In this case, $l(\theta, x)$ is solely determined by $x$, so that we always have $\nabla_{\theta} \\ l(\theta, x) = 0$. Hence,  if we directly apply the first approach mentioned in our previous response, $\theta$ would always remain unchanged.
> > >
> > >
> > > We hope the above two examples are sufficient to show that **Stackelberg games/bilevel programs/MPECs cannot be directly solved by common online learning algorithms**. If the reviewer thinks it is necessary, we would like to add such explanations to our paper.
> > >
> > >
> > > ##### **2. Utility v.s. cost**
> > >
> > > We define atomic games and nonatomic games in terms of utilities (positive reward) and costs (negative reward) simply because it seems that researchers who study atomic/nonatomic games prefer to use the two terms, respectively. But now we are discussing them together, **we agree that sticking to one term (either utility or cost) would improve the clarity**. We will fix it upon acceptance.

---

> > > > ### Comment · Reviewer_ZpdR · 2022-08-07
> > > > **Thank you**
> > > >
> > > > The examples are helpful, and I would strongly recommend referencing it in the main paper.

---

### Official Review · Reviewer_7JdK · 2022-07-09

**Rating:** 6
**Confidence:** 3
**Soundness:** 3 good
**Presentation:** 3 good
**Contribution:** 3 good

**Summary:**

The authors study how to compute the Stackelberg equilibrium in a Stackelberg (bilevel) game where the first mover is a social designer and the second movers are agents that simultaneously best respond to the designed incentives. Different from existing methods that require solving the agents' level equilibrium repeatedly for each update on the designer's strategy, the authors propose a simultaneous design-and-play algorithm that updates strategies on both levels in each update step. The authors show convergence of the new algorithm mainly under conditions that guarantee the uniqueness of the equilibrium and discussed the multiple equilibria case briefly.

**Questions:**

1. So a natural thought on this new approach is flattening the bilevel game into a single-level game and treating the designer as another agent in the network that is connected to all other agents. If we use this flattened view to solve the variational inequalities directly, can it still work under your assumptions, and what are the main differences between your approach and this "brute force" flattened approach?
2. Same as discussed in the weaknesses, is it possible to add quantitative comparisons between your algorithms and other bi-level optimization and MPEC works?

**Limitations:**

As discussed in the weaknesses, the paper seems to be a bit short of information in its quantitative comparisons to the related works.

**Strengths And Weaknesses:**

Strengths:
1. Clear problem statement
2. Clear presentation of the algorithms and the theoretical guarantees of the new algorithms

Weaknesses:
1. The performance bounds on the related works or baselines are not clearly given, making it hard to know the efficiency improvement of the new algorithms. Personally, I would like to see a table showing what are the common assumptions used in this paper and other related works, and whether each of them guarantees convergence and at what convergence rate, adding other comparison metrics like the order of floating-point operations steps would be even better
2. It would be better to have a table in the appendix showing the notations used in the paper

---

> ### Author Response · Authors · 2022-08-02
> **Response to Reviewer 7JdK**
>
> Thanks for providing comments and suggestions. Below are our responses to your questions.
>
>
> **1. Can we flatten a bilevel game into a single-level game?**
>
> The incentive design problem is formulated as a bilevel program (a Stackelberg game) because the designer has the authority to anticipate and influence the response of the players. In other words, the designer and the players have a typical **leader-follower relationship**.
>
> Of course, the leader would become a player in the lower-level game as long as it **gives up "moving first"**. Then the Stackelberg game would be "**flattened**", using your words, and become a common single-level game. The resulting game is much easier to solve, but this transformation is not equivalent. Specifically, the leader’s cost will increase after it decides to compete with the followers on an equal basis, or “a la Cournot”, using the terminology in economics [1].
>
> **2. Comparing complexities with benchmarks.**
>
> To the best of our knowledge, most algorithms proposed for MPECs are purely empirical. But in recent optimization/ML literature, the following two papers [2, 3] have established the theoretical convergence rate of algorithms for solving **bilevel programs whose lower level is an unconstrained optimization problem**, which is more restricted.
>
> [2] Ghadimi, Saeed, and Mengdi Wang. "Approximation methods for bilevel programming."
>
> [3] Hong, Mingyi, et al. "A two-timescale framework for bilevel optimization: Complexity analysis and application to actor-critic."
>
> We will include a table for comparison. Here we first note that (1) our assumptions are generally in line with those in [1, 2] but we need to impose some extra assumptions on the lower-level equilibrium problem, which is more complex than unconstrained optimization problems; (2) our convergence rate is slower than [2] because the equilibrium constraints introduce extra difficulties to the overall problem. To stabilize our algorithm, we have to introduce a "**mixing step**", which would slow down the convergence rate. But without this step, the algorithm would diverge (see the experiment in Appendix F.2).
>
>
> **3. Quantitative comparison to other related works.**
>
> In the ML literature, [4] has studied both implicit differentiation and iterative differentiation based methods for MPECs. We will compare our method with [4] as well as its more recent improvement. But as these methods are all double-looped, the efficiency of our method can be anticipated. Nevertheless, to make a clear comparison, we will add more experimental results to our paper.
>
> [4] Li, Jiayang, et al. "End-to-end learning and intervention in games." Advances in Neural Information Processing Systems 33 (2020).

---

### Official Review · Reviewer_1gpF · 2022-07-11

**Rating:** 7
**Confidence:** 2
**Soundness:** 4 excellent
**Presentation:** 3 good
**Contribution:** 3 good

**Summary:**

In large scale incentive design processes, the classical approach is one of estimating behavior, adjusting incentives, evaluating the outcome, repeated until a fixed point is reached. At each step, you may need to wait long enough for behavior to stabilize into an equilibrium or to see enough data for the econometric task at hand.

This work considers an alternative approach in which the behavior gradient is used to estimate how the behavior will change, and hence allowing the algorithm to adjust incentives faster to the way that agents are responding to the incentives, effectively adjusting both in continuous time to lead to equilibrium (e.g., more of an optimal control based approach).

The approach is implemented on a few problems: second-best tolling, and pollution taxes, both showing empirical convergence in line with the theoretical convergence results.


**Questions:**

Can you discuss the improvement in theoretical convergence or empirical performance relative to the best known prior approaches?

**Limitations:**

Yes, the authors have adequately addressed limitations.

**Strengths And Weaknesses:**

The work seems to be a substantial original contribution in showing that simultaneous design and react can be used to quickly converge to optimal policies, across a broad range of problems.

The exposition does a good job of clearly illustrating the robustness of the techniques.

The simultaneous design & move approach is advocated for as an improvement over waiting for convergence to equilibrium at each step. While this claim makes intuitive sense that it would be an improvement, it would strengthen the argument to show an explicit theoretical or empirical gain in convergence relative to the bilevel approaches.

---

> ### Author Response · Authors · 2022-08-02
> **Response to Reviewer 1gpF**
>
> Thanks for your positive view of our work. Below we will make a comparison between our work and previous work.
>
> ##### **1. Theoretically**
>
> In recent optimization/ML literature, the following two papers are often recognized as benchmarks for bilevel programming.
>
> [1] Ghadimi, Saeed, and Mengdi Wang. "Approximation methods for bilevel programming."
>
> [2] Hong, Mingyi, et al. "A two-timescale framework for bilevel optimization: Complexity analysis and application to actor-critic."
>
> Both of them propose algorithms for solving bilevel programs and **theoretically establish the convergence rate**.
>
> However, their algorithms and analysis are restricted to **bilevel programs in which the lower level is an unconstrained optimization problem**. Instead, the incentive design problem studied in our paper is formulated as an MPEC (mathematical program with equilibrium constraints). Their methods cannot be directly extended to MPECs. Hence, we think it is not fair to compare the convergence rate of our method with theirs.
>
> To the best of our knowledge, **no previous work** has established the theoretical convergence rate of any numerical algorithms for MPECs recently. So, we hope our paper can become a benchmark like [1, 2] for MPECs.
>
>
> ##### **2. Empirically**
>
> Most algorithms proposed for MPECs are purely empirical. To the best of our knowledge, all of these methods are double-loop. We have conducted many experiments to compare these algorithms [e.g., the two methods proposed in 3] with our method before. In fact, it is the inefficiency of these methods that has motivated us to devise new algorithms. Thanks to the single-loop structure of our new algorithm, it takes much less running time and we would like to include these results for better illustration.
>
> [3] Li, Jiayang, et al. "End-to-end learning and intervention in games." Advances in Neural Information Processing Systems 33 (2020).

---

### Meta-Review · Area_Chair_hUn4 · 2022-08-29

**Recommendation:** Accept
**Confidence:** Certain

**Metareview:**

Executive summary:

This paper considers the incentive design problem (making sure players in a non-cooperative game play a certain equilibrium by setting incentives appropriately) in both atomic and non-atomic games. It deviates from the classic computational learning approach which is looping over two steps: (1) setting incentives and (2) waiting for the system to converge to equilibrium. This work proposes a "simultaneous design and play" approach in which the behavior gradient is used to estimate how the behavior will change, and hence allowing the algorithm to adjust incentives faster to the way that agents are responding to the incentives, effectively adjusting both in continuous time to lead to equilibrium.

On the theoretical side, the paper establishes that this "single loop" approach provably converges in two important cases (Theorem 5.7 and Theorem 5.8). On the empirical side, the paper applies the proposed approach to two problems (second-best tolling, and pollution taxes).

Discussion and recommendation:

We had a very lively discussion around this paper. All reviewers agreed that the paper pursues a natural and original idea, but all reviewers had important objections and/or suggestions.

Our conclusion after the rebuttals was that all these can be addressed through proper rewriting.

We in particular felt that it is crucial to corroborate the intuitive claim that the simultaneous approach is more efficient with numerical results. So please include a numerical comparison of your proposed approach to state-of-the-art algorithms using the two step approach as e.g. promised in the reply to reviewer 1gpF.

Weak accept.

**Award:**

No

---

### Decision · Program_Chairs · 2022-09-14

Accept